# RIME: Robust Preference-based Reinforcement Learning with Noisy Human Preferences

## Abstract

Designing an effective reward function remains a significant challenge in numerous reinforcement learning (RL) applications. Preference-based Reinforcement Learning (PbRL) presents a novel framework that circumvents the need for reward engineering by harnessing human preferences as the reward signal. However, current PbRL algorithms primarily focus on feedback efficiency, which heavily depends on high-quality feedback from domain experts. This over-reliance results in a lack of robustness, leading to a severe performance degradation under noisy feedback conditions, thereby limiting the broad applicability of PbRL. In this paper, we present RIME, a robust PbRL algorithm for effective reward learning from noisy human preferences. Our method incorporates a sample selection-based discriminator to dynamically filter denoised preferences for robust training. To mitigate the accumulated error caused by incorrect selection, we propose to warm start the reward model for a good initialization, which additionally bridges the performance gap during transition from pre-training to online training in PbRL. Our experiments on robotic manipulation and locomotion tasks demonstrate that RIME significantly enhances the robustness of the current state-of-the-art PbRL method. Ablation studies further demonstrate that the warm start is crucial for both robustness and feedback-efficiency in limited-feedback cases.

## 1 Introduction

Reinforcement Learning (RL) has demonstrated remarkable performance in various domains, including gameplay (Vinyals et al., 2019; Perolat et al., 2022; Kaufmann et al., 2023), robotics (Chen et al., 2022), autonomous systems (Bellemare et al., 2020; Zhou et al., 2020), etc. However, the key determinant of RL success often hinges on the careful design of reward functions, which can be both labor-intensive and error-prone. In this context, Preference-Based RL (PbRL) (Akrour et al., 2011; Christiano et al., 2017) emerges as a valuable alternative, negating the need for hand-crafted reward functions. PbRL adopts a human-in-the-loop paradigm, where human teachers provide preferences over distinct agent behaviors as the reward signal.

Nevertheless, existing works in PbRL have primarily focused on enhancing feedback efficiency, aiming to maximize the expected return with few number of preference queries. This focus induces a substantial reliance on high-quality human feedback, typically assuming expertise on the human teacher (Liu et al., 2022; Kim et al., 2022). However, humans are prone to errors (Christiano et al., 2017). In broader applications, feedback is often sourced from non-expert users or crowd-sourcing platforms, where the quality can be inconsistent and noisy. Further complicating the matter, Lee et al. (2021a) showed that even a mere 10% corruption rate in preference labels can dramatically degrade the algorithmic performance. The lack of robustness to noisy preference labels hinders the wide application of PbRL.

Meanwhile, learning from noisy labels, also known as robust training, is a rising concern in deep learning, as such labels severely degrade the generalization performance of deep neural networks. Song et al. (2022) classifies current methods for robust training into four key categories: robust architecture (Cheng et al., 2020), robust regularization (Xia et al., 2020), robust loss design (Lyu & Tsang, 2019), and sample selection (Li et al., 2020; Song et al., 2021). However, it poses challenges to incorporate these advanced methods for robust training in PbRL. This complexity arises due to the limited number of feedback, which is often restricted to only hundreds or thousands of feedback in

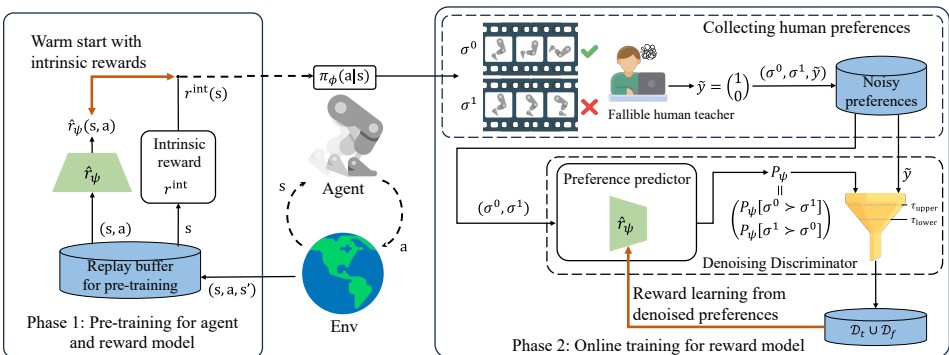

Figure 1: Overview of RIME. In the pre-training phase, we warm start the reward model $\hat{r}_\psi$ with intrinsic rewards $r^{int}$ to facilitate a smooth transition to online training phase. Post pre-training, the policy, Q-network, and reward model $\hat{r}_\psi$ are all inherited as initial configurations for online training. During online training, we utilize a denoising discriminator to screen denoised preferences for robust reward learning. This discriminator employs a dynamic lower bound $\tau_{lower}$ on the KL divergence between predicted preferences $P_\psi$ and annotated preference labels $\tilde{y}$ to filter trustworthy samples $\mathcal{D}_t$, and an upper bound $\tau_{upper}$ to flip highly unreliable labels $\mathcal{D}_f$.

some tasks for the sake of feedback-efficiency and cost reduction, as well as the potential distribution shift issue during RL training.

In this work, we aim to improve the robustness of preference-based RL algorithms on noisy and quantitatively limited human preferences. To this end, we present RIME: **R**obust preference-based re**I**nforcement learning via war**M**-start d**E**noising discriminator. RIME modifies the training paradigm of the reward model in widely-adopted two-phase (i.e., pre-training and online training phases) pipeline of PbRL. Figure 1 shows an overview of RIME. In particular, to empower robust learning from noisy preferences, we introduce a denoising discriminator. This discriminator utilizes dynamic lower and predefined upper bounds on the Kullback–Leibler (KL) divergence between predicted and annotated preference labels to filter denoised samples. Further, to mitigate the accumulated error caused by incorrect filtration, we present to warm start the reward model during the pre-training phase for a good initialization. Moreover, we find that the warm start also bridges the performance gap that occurs during the transition from pre-training to online training

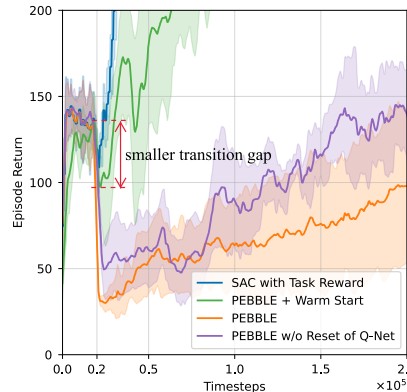

Figure 2: Performance degradation during transition on Walker-walk with 30% noisy preferences. We pre-train a policy and Q-network for 20000 steps.

(Figure 2). This gap is clearly observed under noisy feedback settings and is fatal to robustness. The issue arises because the biased reward model, trained on noisy preferences, biasedly optimize the Q-network through minimizing Bellman residual. This, in turn, offers a poor learning signal for the policy, erasing any gains made during pre-training. Our experiments demonstrate that RIME exceeds existing baselines by a large margin in noisy conditions and considerably improves robustness.

Our contributions can be summarized as follows:

○ We present RIME, a robust algorithm for PbRL, designed to effectively train reward models from noisy feedback—an important and realistic topic that has not been studied extensively.

○ We propose a warm start method to bridge the performance gap during the transition from pre-training to online training in PbRL, which proves to be crucial for both robustness and feedback-efficiency in limited-feedback cases.

○ We demonstrate that RIME outperforms existing PbRL baselines under noisy feedback settings, across a diverse set of robotic manipulation tasks from Meta-World (Yu et al., 2020) and locomotion tasks from the DeepMind Control Suite (Tassa et al., 2018; 2020).

## 2 RELATED WORK

**Preference-based Reinforcement Learning**. The paradigm that incorporates human feedback into the training of reward models has proven effective in various domains, including natural language processing (Ouyang et al., 2022), multi-modal (Lee et al., 2023), and reinforcement learning (Christiano et al., 2017; Ibarz et al., 2018; Hejna III & Sadigh, 2023). In the context of RL, Christiano et al. (2017) proposed a comprehensive framework for PbRL. To improve feedback-efficiency, PEBBLE (Lee et al., 2021b) used unsupervised exploration for policy pre-training. SURF (Park et al., 2021) employed data augmentation and semi-supervised learning to enrich the preference dataset. RUNE (Liang et al., 2021) encouraged exploration by modulating reward uncertainty. MRN (Liu et al., 2022) introduced a bi-level optimization aimed at optimizing the Q-function's performance, yielding improvement in feedback efficiency. PT (Kim et al., 2022) utilized Transformer architecture to model non-Markovian rewards, showing effectiveness in complex tasks.

Despite these advancements, the focus on feedback efficiency should not overshadow the equally critical issue of robustness in PbRL. Lee et al. (2021a) indicated that a mere 10% rate of corrupted preference labels can significantly impair algorithmic performance. Moreover, in broader application scenarios, the gathering of non-expert preferences exacerbates the risk of introducing erroneous labels. Therefore, enhancing the robustness in PbRL remains a vital research direction. In this work, we introduce a denoising discriminator for filtering denoised preferences and a warm start method which is beneficial for both robustness and feedback-efficiency.

**Learning from Noisy Labels**. Learning from noisy labels has gained more attention in supervised learning, particularly due to the prevalence of noisy or imprecise labels in real-world applications. A variety of approaches have been proposed for robust training (Song et al., 2022), including architectural modifications (Goldberger & Ben-Reuven, 2016), regularization (Lukasik et al., 2020), loss function designs (Zhang & Sabuncu, 2018), and sample selection methods (Yu et al., 2019; Nguyen et al., 2019; Wang et al., 2021). In PbRL, Xue et al. (2023) proposed an encoder-decoder architecture to model diverse human preferences, which required huge amount of preference labels (approximate 100 times the amount used in our experiments). Our approach can be situated within the sample selection category and improves robustness while preserving feedback-efficiency.

**Policy-to-Value Reincarnating RL.** Policy-to-value reincarnating RL (PVRL) means transferring a suboptimal teacher policy to a value-based RL student agent (Agarwal et al., 2022). Uchendu et al. (2023) found that a randomly initialized $Q$-network in PVRL leads to the teacher policy being forgotten quickly. Within the widely-adopted pipeline of PbRL, the challenge intrinsic to PVRL also arise during the transition from pre-training to online training, but has been neglected in previous research (Lee et al., 2021b; Park et al., 2021; Liang et al., 2021; Liu et al., 2022). The performance degradation during transition becomes notably pronounced under noisy feedback conditions. Based on this observation, we propose to warm start the reward model for a seamless transition. Our ablation study demonstrates that the warm start is crucial for both robustness and feedback-efficiency.

## 3 PRELIMINARIES

**Preference-based Reinforcement Learning**. In standard RL, an agent interacts with an environment in discrete time steps (Sutton & Barto, 2018). At each time step $t$, the agent observes the current state $\mathbf{s}_t$ and selects an action $\mathbf{a}_t$ according to its policy $\pi(\mathbf{a}_t|\mathbf{s}_t)$. The environment responds by emitting a reward $r(\mathbf{s}_t, \mathbf{a}_t)$ and transitioning to the next state $\mathbf{s}_{t+1}$. The agent's objective is to learn a policy that maximizes the expected return.

In Preference-based RL, there is no predefined reward function. Instead, a teacher offers preferences between agent's behaviors and an estimated reward function $\hat{r}_\psi$ is trained to align with collected preferences. Following previous works (Lee et al., 2021b; Liu et al., 2022; Kim et al., 2022), we consider preferences over two trajectory segments of length $H$, where segment $\sigma = \{(\mathbf{s}_1, \mathbf{a}_1), ..., (\mathbf{s}_H, \mathbf{a}_H)\}$. Given a pair of segments $(\sigma^0, \sigma^1)$, a teacher provides a preference label $\tilde{y}$ from the set $\{(1, 0), (0, 1), (0.5, 0.5)\}$. The label $\tilde{y} = (1, 0)$ signifies $\sigma^0 \succ \sigma^1$, $\tilde{y} = (0, 1)$ signifies $\sigma^1 \succ \sigma^0$, and $\tilde{y} = (0.5, 0.5)$ represents an equally preferable case, where $\sigma^i \succ \sigma^j$ denotes that segment $i$ is preferred over segment $j$. Each feedback is stored in a dataset $\mathcal{D}$ as a triple $(\sigma^0, \sigma^1, \tilde{y})$. Following the Bradley-Terry model (Bradley & Terry, 1952), the preference predicted by the esti-

mated reward function $\hat{r}_\psi$ is formulated as:

$$P_\psi[\sigma^i \succ \sigma^j] = \frac{\exp\left(\sum_t \hat{r}_\psi(\mathbf{s}_t^i, \mathbf{a}_t^i)\right)}{\exp\left(\sum_t \hat{r}_\psi(\mathbf{s}_t^i, \mathbf{a}_t^i)\right) + \exp\left(\sum_t \hat{r}_\psi(\mathbf{s}_t^j, \mathbf{a}_t^j)\right)}. \tag{1}$$

The estimated reward function $\hat{r}_\psi$ is updated by minimizing the cross-entropy loss between the predicted preferences $P_\psi$ and the annotated labels $\tilde{y}$:

$$\mathcal{L}^{\text{CE}}(\psi) = \mathbb{E}\left[\mathcal{L}^{\text{Reward}}\right] = -\mathop{\mathbb{E}}_{(\sigma^0, \sigma^1, \tilde{y}) \sim \mathcal{D}}\left[\tilde{y}(0)\ln P_\psi[\sigma^0 \succ \sigma^1] + \tilde{y}(1)\ln P_\psi[\sigma^1 \succ \sigma^0]\right]. \tag{2}$$

The policy $\pi$ can subsequently be updated using any RL algorithm to maximize the expected return with respect to the estimated reward function $\hat{r}_\psi$.

**Unsupervised Pre-training in PbRL**. Pre-training agents is important in PbRL because the initial random policy often results in uninstructive preference queries, requiring lots of queries for even elementary learning progress. Recent study addressed this issue through unsupervised exploration for policy pre-training (Lee et al., 2021b). Specifically, agents are encouraged to traverse a more expansive state space by utilizing an intrinsic reward function derived from particle-based state entropy (Singh et al., 2003). Formally, the intrinsic reward is defined as (Liu & Abbeel, 2021):

$$r^{\text{int}}(\mathbf{s}_t) = \log(\|\mathbf{s}_t - \mathbf{s}_t^k\|) \tag{3}$$

where $\mathbf{s}_t^k$ is the $k$-th nearest neighbor of $\mathbf{s}_t$. This reward motivates the agent to explore a broader diversity of states. This exploration, in turn, leads to a varied set of agent's behaviors, facilitating more informative preference queries.

**Noisy Human Preferences in PbRL.** Let $\tilde{y}$ denote the annotated preference label and $y$ the ground-truth preference label that is typically sourced from expert human or scripted teachers. Lee et al. (2021a) proposed several noisy labeling models. Among them, the mistake model was the most detrimental to performance across all tested environments, while other noise models even improved performance in most environments. Given these findings, our study specifically addresses robust reward learning from noisy preferences under the mistake model settings. This noise model assumes that the preference dataset is contaminated with corrupted preferences whose annotated labels are $\tilde{y} = (0, 1)$ when $y = (1, 0)$, or $\tilde{y} = (1, 0)$ when $y = (0, 1)$.

## 4 METHOD

In this section, we formally introduce RIME: **R**obust preference-based re**I**nforcement learning via war**M**-start d**E**noising discriminator. RIME consists two main components: 1) a denoising discriminator designed to filter out corrupted preference data while accounting for training instability and distribution shift issue, and 2) a warm start method to effectively initialize the reward model and enable a seamless transition from pre-training to online training. See Figure 1 for the overview of RIME. The full procedure of RIME is detailed in Appendix A.

### 4.1 DENOISING DISCRIMINATOR

In the presence of noisy labels, it is well motivated to distinguish between clean and corrupted samples for robust training. Existing research indicates that deep neural networks first learn generalizable patterns before overfitting to the noise in the data. Therefore, prioritizing samples associated with smaller losses as clean ones is a well-founded approach to improve robustness. Inspired by this insight, we theoretically establish a lower bound on the KL divergence between the predicted preference $P_\psi$ and the annotated preference $\tilde{y}$ for corrupted samples, in order to filter out large-loss corrupted samples.

**Theorem 1** (KL Divergence Lower Bound for Corrupted Samples). *Consider a preference dataset* $\{(\sigma_i^0, \sigma_i^1, \tilde{y}_i)\}_{i=1}^n$, *where* $\tilde{y}_i$ *is the annotated label for the segment pair* $(\sigma_i^0, \sigma_i^1)$ *with the ground truth label* $y_i$. *Let* $x_i$ *denote the tuple* $(\sigma_i^0, \sigma_i^1)$. *Assume the cross-entropy loss* $\mathcal{L}^{\text{CE}}$ *for clean data (whose* $\tilde{y}_i = y_i$*) is bounded by* $\rho$. *Then, the KL divergence between the predicted preference* $P_\psi(x)$ *and the annotated label* $\tilde{y}(x)$ *for a corrupted sample* $x$ *is lower-bounded as follows:*

$$D_{\text{KL}}\left(P_\psi(x)\|\tilde{y}(x)\right) \geq -\ln\rho + \frac{\rho}{2} + \mathcal{O}(\rho^2) \tag{4}$$

The proof of Theorem 1 is presented in Appendix B. Based on Theorem 1, we formulate the lower bound on KL divergence threshold to filter out untrustworthy samples as $\tau_{\text{base}} = -\ln \rho + \alpha \rho$ in practical, where $\rho$ denotes the maximum cross-entropy loss on trustworthy samples observed during the last update, and $\alpha$ serves as a tunable hyperparameter with a value range in $(0, 0.5]$ in theoretical.

However, the shifting state distribution complicates the robust training problem in RL, compared to deep learning contexts. To add tolerance for trustworthy samples in cases of distribution shift, we introduce an auxiliary term characterizing the uncertainty for filtration, defined as $\tau_{\text{unc}} = \beta_t \cdot s_{\text{KL}}$, where $\beta_t$ is a time-dependent parameter, and $s_{\text{KL}}$ is the standard deviation of the KL divergence. Our intuition is that the inclusion of out-of-distribution data for training is likely to induce fluctuations in the training loss. Therefore, the complete threshold equation is formulated as follows:

$$\tau_{\text{lower}} = \tau_{\text{base}} + \tau_{\text{unc}} = -\ln \rho + \alpha \rho + \beta_t \cdot s_{\text{KL}} \tag{5}$$

We utilize a linear decay schedule for $\beta_t$ to initially allow greater tolerance for samples while becoming increasingly conservative over time, i.e., $\beta_t = \max(\beta_{\min}, \beta_{\max} - kt)$. At each training step for the reward model, we apply the threshold in Equation (5) to identify trustworthy sample dataset $\mathcal{D}_t$, as described below:

$$\mathcal{D}_t = \{(\sigma^0, \sigma^1, \tilde{y}) \mid D_{\text{KL}}(P_\psi(\sigma^0, \sigma^1) \| \tilde{y}) < \tau_{\text{lower}}\} \tag{6}$$

To ensure efficient usage of samples, we introduce a label-flipping method for the reintegration of untrustworthy samples. Specifically, we pre-define an upper bound $\tau_{\text{upper}}$ and reverse the labels for samples exceeding this threshold:

$$\mathcal{D}_f = \{(\sigma^0, \sigma^1, 1 - \tilde{y}) \mid D_{\text{KL}}(P_\psi(\sigma^0, \sigma^1) \| \tilde{y}) > \tau_{\text{upper}}\} \tag{7}$$

Beyond improving sample utilization, the label-flipping method also bolsters the model's predictive confidence and reduce output entropy (Grandvalet & Bengio, 2004). Following two filtering steps, the reward model is trained on the unified datasets $\mathcal{D}_t \cup \mathcal{D}_f$, using the loss function in Equation (8).

$$\mathcal{L}^{\text{CE}} = \mathop{\mathbb{E}}_{(\sigma^0, \sigma^1, \tilde{y}) \sim \mathcal{D}_t} \left[ \mathcal{L}^{\text{Reward}}(\sigma^0, \sigma^1, \tilde{y}) \right] + \mathop{\mathbb{E}}_{(\sigma^0, \sigma^1, 1 - \tilde{y}) \sim \mathcal{D}_f} \left[ \mathcal{L}^{\text{Reward}}(\sigma^0, \sigma^1, 1 - \tilde{y}) \right] \tag{8}$$

Our denoising discriminator belongs to the category of sample selection methods for robust training (Song et al., 2022). However, it is different in using a dynamically adjusted threshold, augmented by a term that captures distributional shifts, making it more suitable for RL training process.

## 4.2 WARM START

Sample selection methods frequently suffer from accumulated error due to incorrect selection, which underscores the need of good initialization for the denoising discriminator to effectively differentiate between samples at initial. Meanwhile, we observe a marked degradation of performance during transition from pre-training to online training (Figure 2). This issue is exacerbated when following the most widely-adopted backbone algorithm, PEBBLE, which resets the Q-network and only retains the pre-trained policy after the pre-training phase.

Inspired by these observations, we propose to warm start the reward model to facilitate a smoother transition from pre-training to online training. Specifically, we pre-train the reward model to approximate intrinsic rewards during pre-training phase. Because the output layer of the reward model typically uses the tanh activation function (Lee et al., 2021b), we firstly normalize the intrinsic reward to the range $(-1, 1)$ as follows:

$$r_{\text{norm}}^{\text{int}}(\mathbf{s}_t) = \text{clip}\left(\frac{r^{\text{int}}(\mathbf{s}_t) - \bar{r}}{3\sigma_r}, -1 + \delta, 1 - \delta\right) \tag{9}$$

where $0 < \delta \ll 1$. $\bar{r}$ and $\sigma_r$ represent the mean and standard deviation of the intrinsic rewards, respectively. Then the agent receives the reward $r_{\text{norm}}^{\text{int}}$ and stores each tuple $(\mathbf{s}_t, \mathbf{a}_t, r_{\text{norm}}^{\text{int}}, \mathbf{s}_{t+1})$ in a replay buffer, denoted as $\mathcal{D}_{\text{pretrain}}$. During the reward model update, we sample batches of $(\mathbf{s}_t, \mathbf{a}_t)$ along with all encountered states $\mathcal{S} = \{\mathbf{s} | \mathbf{s} \text{ in } \mathcal{D}_{\text{pretrain}}\}$ for nearest neighbor searches. The loss function for updating the reward model $\hat{r}_\psi$ is given by the mean squared error as:

$$\mathcal{L}^{\text{MSE}} = \mathop{\mathbb{E}}_{(\mathbf{s}_t, \mathbf{a}_t) \sim \mathcal{D}_{\text{pretrain}}} \left[ \frac{1}{2} \left( \hat{r}_\psi(\mathbf{s}_t, \mathbf{a}_t) - r_{\text{norm}}^{\text{int}}(\mathbf{s}_t) \right)^2 \right] \tag{10}$$

Attributed to warm start, both the Q-network and reward model are aligned with intrinsic rewards, allowing for the retention of all knowledge gained during pre-training (i.e., policy, Q-network, and reward model) for subsequent online training. Moreover, the warm-started reward model contains more information than random initialization, enhancing the discriminator's ability at initial.

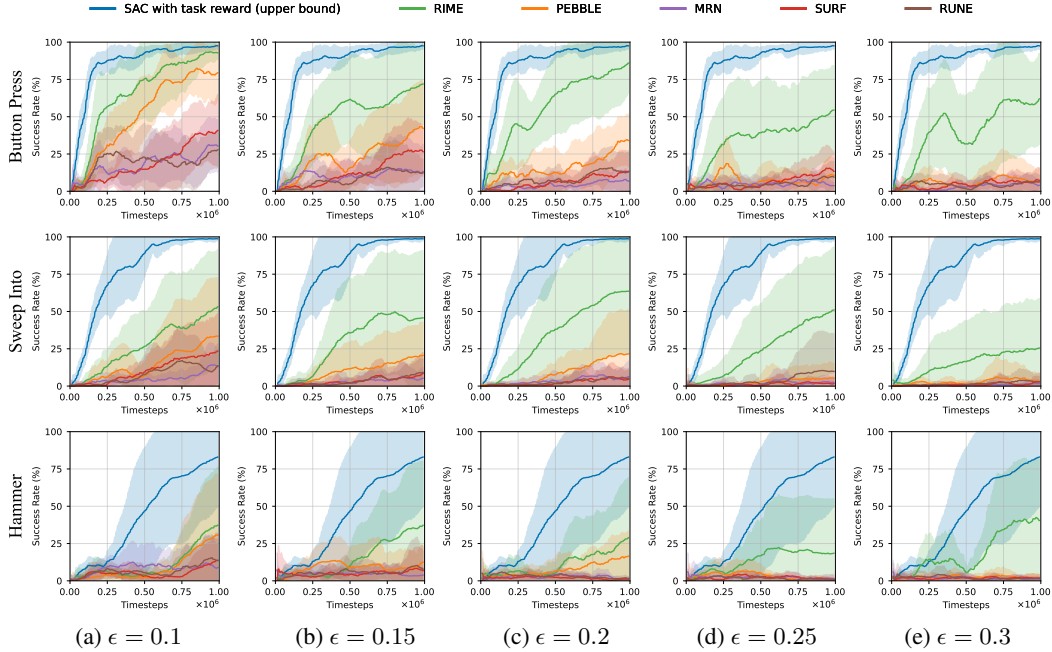

Figure 3: Learning curves for robotic manipulation tasks from Meta-world, where each row represents a specific task and each column corresponds to a different error rate $\epsilon$. SAC serves as a performance upper bound, using a ground-truth reward function unavailable in PbRL settings. The corresponding number of feedback in total and per session are show in Table 8. The solid line and shaded regions respectively denote mean and standard deviation of success rate, across ten runs.

## 5 EXPERIMENTS

We design our experiments to investigate the following:

- How does RIME improve the existing preference-based RL methods in terms of robustness?
- How does RIME perform on clean preferences compared with feedback-efficient baselines?
- What is the contribution of each of the proposed components in RIME?
- Is denoising discriminator better than existing sample selection methods in terms of robustness for reward learning?

### 5.1 SETUPS

We evaluate RIME on a total of six complex tasks, including robotic manipulation tasks from Meta-world (Yu et al., 2020) and locomotion tasks from DMControl (Tassa et al., 2018; 2020). The details of experimental tasks are shown in Appendix C.1. Similar to prior works (Lee et al., 2021a;b; Park et al., 2021), to ensure a systematic and fair evaluation, we consider a scripted teacher that provides preferences between two trajectory segments based on the sum of ground-truth reward values for each segment. To generate noisy preferences, we follow the procedure of the mistake model in Lee et al. (2021a), which flips correct preferences with a probability of $\epsilon$. We refer to $\epsilon$ as the error rate. We choose PEBBLE (Lee et al., 2021b) as our backbone algorithm to implement RIME. In our experiments, we compare RIME against ground-truth reward-based SAC and four state-of-the-art PbRL algorithms: PEBBLE (Lee et al., 2021b), SURF (Park et al., 2021), RUNE (Liang et al., 2021), and MRN (Liu et al., 2022). Here, SAC is considered as an upper bound for performance, as it utilizes a ground-truth reward function not available in PbRL settings. We include SAC in our comparisons because it is the backbone RL algorithm of PEBBLE.

**Implementation Details.** For all experiments, we use the same hyperparameters used by PEBBLE algorithm, which are specified in Appendix C.2. For query selection strategy, we use the uniform sampling scheme. For the implementation of baselines, we use their corresponding publicly released

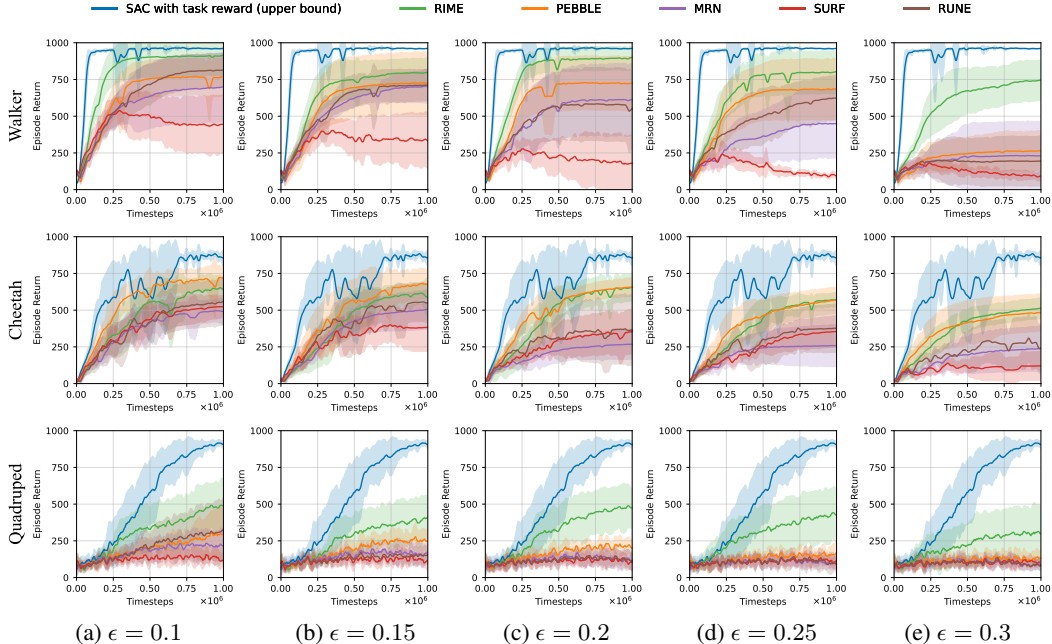

Figure 4: Learning curves on locomotion tasks from DMControl, where each row represents a specific task and each column corresponds to a different error rate $\epsilon$ setting. SAC serves as a performance upper bound, using a ground-truth reward function unavailable in PbRL settings. The corresponding number of feedback in total and per session are show in Table 8. The solid line and shaded regions respectively denote mean and standard deviation of episode return, across ten runs.

repositories and keep the hyperparameters consistent with their original configurations (see Table 1 for source codes). For the hyperparameters of RIME, we fix $\alpha = 0.5$, $\beta_{\min} = 1$ and $\beta_{\max} = 3$ in the lower bound $\tau_{\text{lower}}$, and fix the upper bound $\tau_{\text{upper}} = 3\ln(10)$ for all experiments. The decay rate $k$ is $1/30$ for tasks from DMControl, and $1/300$ for tasks from Meta-world, respectively. The total feedback amount and feedback amount per session in each condition are detailed in Table 8.

For each task, we run all algorithms independently for ten times and report the average performance along with the standard deviation. Tasks from Meta-world are measured on success rate, while tasks from DMControl are measured on ground-truth episode return. More details on network architectures and hyperparameters are provided in Appendix C.2.

## 5.2 RESULTS

**Meta-world Tasks.** For robotic manipulation tasks, we consider three tasks from Meta-world: Button-press, Sweep-into, and Hammer, to investigate how RIME improves robustness of a PbRL algorithm. The examples and details of tasks are available in Appendix C.1.

Figure 3 shows the learning curves of RIME and baselines on Meta-world tasks across five error rates $\epsilon \in \{0.1, 0.15, 0.2, 0.25, 0.3\}$. In the same condition, defined by both the environment and the error rate $\epsilon$, algorithms use an identical number of preference queries for fair comparison. As shown in Figure 3, RIME exceeds the PbRL baselines by a large margin for all evaluated conditions. Specifically, RIME remains effective (approaching or exceeding 50% success rate) in conditions where all baselines struggle, including Button-press with $\epsilon \geq 0.2$, Sweep-into with $\epsilon = 0.2$ and $0.25$, and Hammer with $\epsilon = 0.3$. These results demonstrate that RIME significantly improves robustness against noisy preferences. We also observe that baselines does not work in most noisy conditions on Meta-world tasks. The reason is that both the difficulty of Meta-world tasks and the pursuit of feedback efficiency of baselines lead to over-reliance on feedback quality.

**DMControl Tasks.** For locomotion tasks, we choose three complex environments from DMControl: Walker-walk, Cheetah-run, and Quadruped-walk. Figure 4 shows the learning curves for all considered algorithms on DMControl tasks across five distinct error rates. RIME still shows obvious

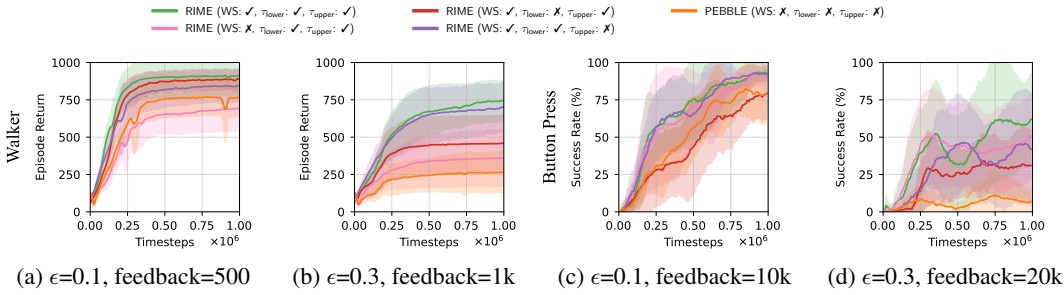

Figure 5: Ablation study of components in RIME, including warm start (WS), lower bound $\tau_{\text{lower}}$, and upper bound $\tau_{\text{upper}}$, on Walker-walk (a and b) and Button-press (c and d) with different error rate $\epsilon \in \{0.1, 0.3\}$, across five runs.

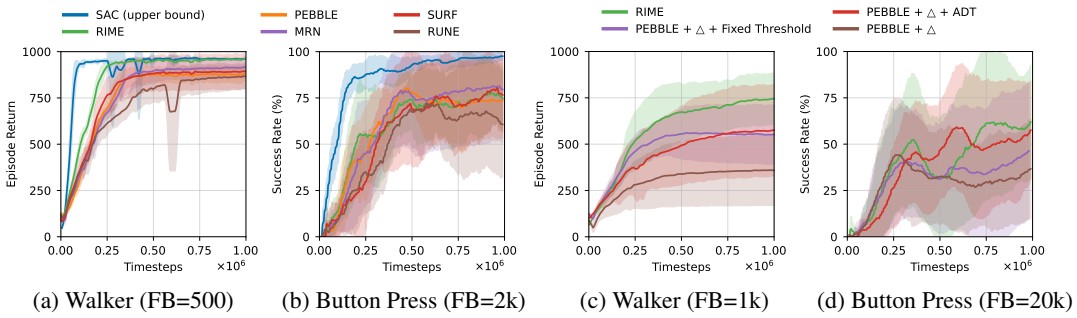

Figure 6: Ablation study on 1: clean preference data (a and b), 2: comparison with other sample selection methods for robust training with feedback error rate $\epsilon = 0.3$ (c and d, where the symbol $\triangle$ means warm start + label flipping). FB means feedback. The results show the mean and standard deviation averaged over ten runs.

advantages compared with baselines. We also observe that RIME is the only algorithm that successfully trains effective agents on Walker-walk using only 1000 feedback with an error rate of $\epsilon = 0.3$. ~~Given that it takes only few human minutes to provide 1000 preference labels (Park et al., 2021), RIME improves robustness while still maintaining feedback efficiency.~~ These results again demonstrate that RIME improves robustness of the PbRL method on a variety of complex tasks. Additionally, we find that baselines perform relatively better on DMControl than on Meta-world tasks under noisy setttings, but still almost fail on the hard task, Quadruped, when the error rate $\epsilon \geq 0.2$. This observation suggests that as task complexity increases, there is a commensurate rise in the requirement for high-quality human feedback.

## 5.3 ABLATION STUDY

**Component analysis.** We perform ablation study to individually evaluate each technique in RIME: warm start (WS), lower bound $\tau_{\text{lower}}$, and upper bound $\tau_{\text{upper}}$ of KL divergence. We present results in Figure 5 in which we compare the performance of removing each component from RIME. We observe that warm start is crucial for robustness when the number of feedback is quite limited (Figure 5a and 5b). This is because the limited samples restrict the capability of the reward model, leading to more rounds of queries to cross the transition gap. Moreover, it might be hard to distinguish for discriminator at initial with limited samples, which urges for good initialized reward models.

The lower bound $\tau_{\text{lower}}$ for filtering trustworthy samples is important in high error rate (Figure 5b and 5d) and adequate feedback (Figure 5c and 5d) situations. The upper bound $\tau_{\text{upper}}$ for flipping labels always brings some performance improvements in our ablation experiments. The full algorithm outperforms every other combination in most tasks. Additionally, the results show that although the contribution of warm start and denoising discriminator vary in different environments at lower error rates, they are both effective and their combination proves essential for the overall success of our method in environments with high error rates.

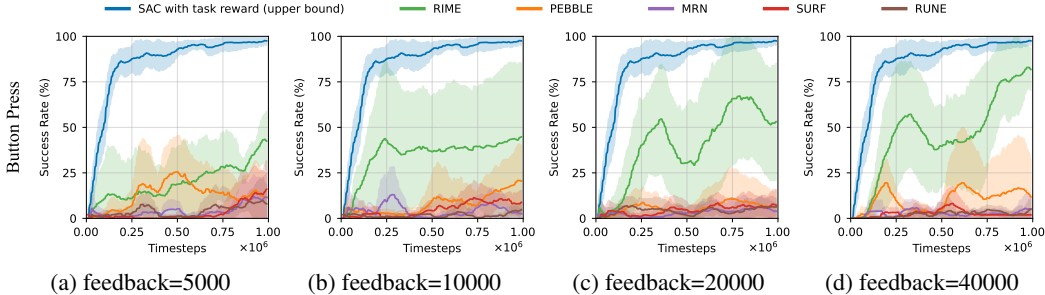

Figure 7: Ablation study on the effects of feedback volume on Button-press, where error rate is fixed at 0.3. The results show the mean and standard deviation averaged over five runs.

**Performance on clean preferences.** A robust algorithm should not perform poorly in non-noisy environments. To investigate this, we perform ablation study using clean preference data on Walker-walk and Button-press with 500 and 2000 feedback, respectively. Notably, we even do not remove the denoising discriminator in RIME for this experiment, which results in that RIME uses fewer correct samples and some corrupted samples from flipping for training. Despite this, as shown in Figure 6a and 6b, we surprisingly find that RIME achieves competitive or even better performance compared with other feedback-efficient baselines on Button-press and Walker-walk, respectively. This result demonstrates that warm start for reward model is also important for feedback-efficiency. This is because it bridges the performance gap during transition from pre-training to online training, thereby saving rounds of queries for subsequent learning process.

**Comparison with other sample selection methods for robust training.** To demonstrate that our denoising discriminator can induce significant improvements on robustness for reward learning, we compare our method with other sample selection methods for robust training. We consider using a fixed threshold and adaptive denoising training (ADT) proposed in Wang et al. (2021) as our baselines. The former method select samples whose cross-entropy loss is less than the threshold as training samples. We set the fixed threshold to the final average values of $\tau_{\text{lower}}$ in RIME. ADT drops a-$\tau(t)$ proportion of samples with the largest cross-entropy loss at each training iteration, where $\tau(t) = \min(\gamma t, \tau_{\text{max}})$. We choose parameters for ADT as $\tau_{\text{max}} = 0.3$, $\gamma = 0.003$ and 0.0003 for tasks from DMControl and Meta-world, respectively. For fair comparision, we add warm start and label flipping for all algorithms in this experiment. As show in Figure 6c and 6d, we observe that both fixed threshold and ADT improves the performance of PEBBLE under high-level noisy conditions, but RIME still outperforms these two methods. This might be because RIME takes consideration of training instability and distribution shift issue into the threshold.

**Effects of feedback volume.** To investigate effects of the number of noisy feedback on the performance of preference-based RL algorithms, we conduct evaluations comparing RIME with existing baselines across a range of query sizes at a fixed error rate of $\epsilon = 0.3$, on Button-press. As illustrated in Figure 7, RIME demonstrates a consistent enhancement in performance with increasing volumes of noisy feedback, while baselines' performance does not change obviously.

## 6 CONCLUSION

In this paper, we present RIME, a robust algorithm for preference-based reinforcement learning (PbRL) designed for effective reward learning from noisy preferences. Unlike previous research which primarily aims to enhance feedback efficiency, RIME focuses on improving robustness by employing a sample selection-based discriminator to dynamically denoise preferences. To reduce accumulated error due to incorrect selection, we utilize a warm-start method for the reward model, enhancing the initial capability of the denoising discriminator. The warm-start approach also serves to bridge the performance gap during transition, facilitating a seamless transition from pre-training to online training. Our experiments show that RIME substantially boosts the robustness of the state-of-the-art PbRL method across a range of complex robotic manipulation and locomotion tasks. Ablation studies further demonstrate that the warm-start approach is crucial for both robustness and feedback efficiency. We believe that RIME has the potential to broaden the applicability of PbRL by leveraging preferences from non-expert users or crowd-sourcing platforms.

**Reproducibility statement.** We describe the implementation details of RIME in Appendix A, and also provide our source code in the supplementary material.

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

# Appendix

## A  RIME ALGORITHM DETAILS

In this section, we provide the full procedure for RIME based on the backbone PbRL algorithm, PEBBLE (Lee et al., 2021b), in Algorithm 1.

---

**Algorithm 1** RIME

---

1: Initialize policy $\pi_\phi$, Q-network $Q_\theta$ and reward model $\hat{r}_\psi$
2: Initialize replay buffer $\mathcal{B} \leftarrow \emptyset$
3: // UNSUPERVISED PRE-TRAINING
4: **for** each pre-training step $t$ **do**
5:     Collect $\mathbf{s}_{t+1}$ by taking $\mathbf{a}_t \sim \pi_\phi(\mathbf{a}_t|\mathbf{s}_t)$
6:     Compute normalized intrinsic reward $r_{\text{norm},t}^{\text{int}} \leftarrow r_{\text{norm}}^{\text{int}}(\mathbf{s}_t)$ as in Equation (9)
7:     Store transitions $\mathcal{B} \leftarrow \mathcal{B} \cup \left\{(\mathbf{s}_t, \mathbf{a}_t, \mathbf{s}_{t+1}, r_{\text{norm},t}^{\text{int}})\right\}$
8:     **for** each gradient step **do**
9:         Sample minibatch $\left\{(\mathbf{s}_j, \mathbf{a}_j, \mathbf{s}_{j+1}, r_{\text{norm},j}^{\text{int}})\right\}_{j=1}^{B} \sim \mathcal{B}$
10:         Optimize policy and Q-network with respect to $\phi$ and $\theta$ using SAC
11:         // WARM START
12:         Update reward model $\hat{r}_\psi$ according to Equation (10)
13:     **end for**
14: **end for**
15: // ONLINE TRAINING
16: Initialize the maximum KL divergence value $\rho = \infty$
17: Initialize a dataset of noisy preferences $\mathcal{D}_{\text{noisy}} \leftarrow \emptyset$
18: **for** each training step $t$ **do**
19:     // ROBUST REWARD LEARNING
20:     **if** $t\%K == 0$ **then**
21:         Generate queries from replay buffer $\{(\sigma_i^0, \sigma_i^1)\}_{i=1}^{N_{\text{query}}} \sim \mathcal{B}$ and corresponding human feedback $\{\tilde{y}_i\}_{i=1}^{N_{\text{query}}}$
22:         Store preferences $\mathcal{D}_{\text{noisy}} \leftarrow \mathcal{D}_{\text{noisy}} \cup \{(\sigma_i^0, \sigma_i^1, \tilde{y}_i)\}_{i=1}^{N_{\text{query}}}$
23:         Compute lower bound $\tau_{\text{lower}}$ according to Equation (5)
24:         Filter trustworthy samples $\mathcal{D}_t$ using lower bound $\tau_{\text{lower}}$ as in Equation (6)
25:         Flip labels using upper bound $\tau_{\text{upper}}$ to obtain dataset $\mathcal{D}_f$ as in Equation (7)
26:         Update reward model $\hat{r}_\psi$ with samples from $\mathcal{D}_t \cup \mathcal{D}_f$ according to Equation (8)
27:         Relabel entire replay buffer $\mathcal{B}$ using $\hat{r}_\psi$
28:         Update parameter $\rho$ with the maximum KL divergence between predicted and annotated labels in dataset $\mathcal{D}_t \cup \mathcal{D}_f$
29:     **end if**
30:     **for** each timestep $t$ **do**
31:         Collect $\mathbf{s}_{t+1}$ by taking $\mathbf{a}_t \sim \pi_\phi(\mathbf{a}_t|\mathbf{s}_t)$
32:         Store transitions $\mathcal{B} \leftarrow \mathcal{B} \cup \{(\mathbf{s}_t, \mathbf{a}_t, \mathbf{s}_{t+1}, \hat{r}_\psi(\mathbf{s}_t, \mathbf{a}_t))\}$
33:     **end for**
34:     **for** each gradient step **do**
35:         Sample minibatch from replay buffer $\{(\mathbf{s}_j, \mathbf{a}_j, \mathbf{s}_{j+1}, \hat{r}_\psi(\mathbf{s}_j, \mathbf{a}_j))\}_{j=1}^{B} \sim \mathcal{B}$
36:         Optimize policy and Q-network with respect to $\phi$ and $\theta$ using SAC
37:     **end for**
38: **end for**

---

## B  PROOFS FOR THEOREM 1

**Theorem 1.** *Consider a preference dataset $\{(\sigma_i^0, \sigma_i^1, \tilde{y}_i)\}_{i=1}^{n}$, where $\tilde{y}_i$ is the annotated label for the segment pair $(\sigma_i^0, \sigma_i^1)$ with the ground truth label $y_i$. Let $x_i$ denote the tuple $(\sigma_i^0, \sigma_i^1)$. Assume the cross-entropy loss $\mathcal{L}^{\text{CE}}$ for clean data (whose $\tilde{y}_i = y_i$) within this distribution is bounded by $\rho$.*

*Then, the KL divergence between the predicted preference $P_\psi(x)$ and the annotated label $\tilde{y}(x)$ for a corrupted sample $x$ is lower-bounded as follows:*

$$D_{\mathrm{KL}}\left(P_\psi(x)\|\tilde{y}(x)\right) \geq -\ln \rho + \frac{\rho}{2} + \mathcal{O}(\rho^2) \tag{11}$$

*Proof.* For a clean sample $(\sigma^0, \sigma^1)$ with annotated label $\tilde{y}$ and ground-truth label $y$, we have $\tilde{y} = y$. Denote the predicted label as $P_\psi$. In PbRL, the value of $y(0)$ can take one of three forms: $y(0) \in \{0, 0.5, 1\}$. We categorize and discuss these situations as follows:

1. For $y(0) = 0$:

   Because the cross-entropy loss $\mathcal{L}^{\mathrm{CE}}$ for clean data is bounded by $\rho$, we can express:

   $$\mathcal{L}^{\mathrm{CE}}(P_\psi, \tilde{y}) = -\ln(1 - P_\psi(0)) \leq \rho \tag{12}$$

   From the above, we have:

   $$P_\psi(0) \leq 1 - \exp(-\rho) \tag{13}$$

   Then if the label is corrupted, denoted by $\tilde{y}_c$ (i.e., $\tilde{y}_c = (1, 0)$ in this case), the KL divergence between predicted label and corrupted label is formulated as follows:

   $$D_{\mathrm{KL}}(P_\psi\|\tilde{y}_c) = -\ln P_\psi(0) \geq -\ln(1 - \exp(-\rho)) \tag{14}$$

2. For $y(0) = 1$:

   The discussion parallels the $y(0) = 0$ case. Hence, the KL divergence between the predicted label and the corrupted label also maintains a lower bound:

   $$D_{\mathrm{KL}}(P_\psi\|\tilde{y}_c) \geq -\ln(1 - \exp(-\rho)) \tag{15}$$

3. For $y(0) = 0.5$:

   Although this case is not under the mistake model settings (Lee et al., 2021a), the lower bound still holds in this case. Due to the bounded cross-entropy loss $\mathcal{L}^{\mathrm{CE}}$ for clean data, we have:

   $$\mathcal{L}^{\mathrm{CE}}(P_\psi, \tilde{y}) = -\frac{1}{2}\ln P_\psi(0) - \frac{1}{2}\ln(1 - P_\psi(0)) \leq \rho \tag{16}$$

   Solving the inequality (16), we can get:

   $$P_\psi(0)^2 - P_\psi(0) + \exp(-2\rho) \leq 0 \tag{17}$$

   When $\rho \geq \ln 2$, the inequality (17) has a solution:

   $$1 - p \leq P_\psi(0) \leq p \tag{18}$$

   where $p = \frac{1+\sqrt{1-4\exp(-2\rho)}}{2}$.

   Then if the label is corrupted, i.e., $\tilde{y}_c \in \{(0, 1), (1, 0)\}$, the KL divergence between predicted label and corrupted label is formulated as follows:

   $$D_{\mathrm{KL}}(P_\psi\|\tilde{y}_c) \geq \min(-\ln P_\psi(0), -\ln(1 - P_\psi(0))) = -\ln p \tag{19}$$

   Construct an equation about $\rho$:

   $$f(\rho) = p - 1 + \exp(-\rho) = \frac{1 + \sqrt{1 - 4\exp(-2\rho)}}{2} - 1 + \exp(-\rho) \tag{20}$$

   where $\rho \geq \ln 2$.

   Denote $z = \exp(-\rho)$, Equation (20) can be simplified as follows:

   $$f(z) = z + \frac{\sqrt{1 - 4z^2}}{2} - \frac{1}{2} \tag{21}$$

where $0 < z \le \frac{1}{2}$.

Derivative of function $f$ with respect to $z$, we have:

$$f'(z) = 1 - 2\sqrt{\frac{1}{\frac{1}{z^2} - 4}} \tag{22}$$

Function $f'(z)$ decreases monotonically when $z \in (0, 0.5]$, is greater than 0 on the interval $(0, \frac{\sqrt{2}}{4})$, and is less than 0 on the interval $(\frac{\sqrt{2}}{4}, 0.5]$. Therefore, we have:

$$f(z) \le \max(f(0), f(\frac{1}{2})) = 0 \tag{23}$$

Thus, $p \le 1 - \exp(-\rho)$ when $\rho \ge \ln 2$. In turn, we have:

$$D_{\mathrm{KL}}(P_\psi \| \tilde{y}_c) = -\ln p \ge -\ln(1 - \exp(-\rho)) \tag{24}$$

To sum up, inequality (25) holds for the corrupted samples:

$$D_{\mathrm{KL}}(P_\psi \| \tilde{y}_c) \ge -\ln(1 - \exp(-\rho)) \tag{25}$$

Perform Taylor expansion of the lower bound at $\rho = 0$, we can get:

$$D_{\mathrm{KL}}(P_\psi \| \tilde{y}_c) \ge -\ln(1 - \exp(-\rho)) = -\ln \rho + \frac{\rho}{2} + \mathcal{O}(\rho^2) \tag{26}$$

$\square$

## C  EXPERIMENTAL DETAILS

### C.1  TASKS

The robotic manipulation tasks from Meta-world (Yu et al., 2020) and locomotion tasks from DM-Control (Tassa et al., 2018; 2020) used in our experiments are shown in Figure 8.

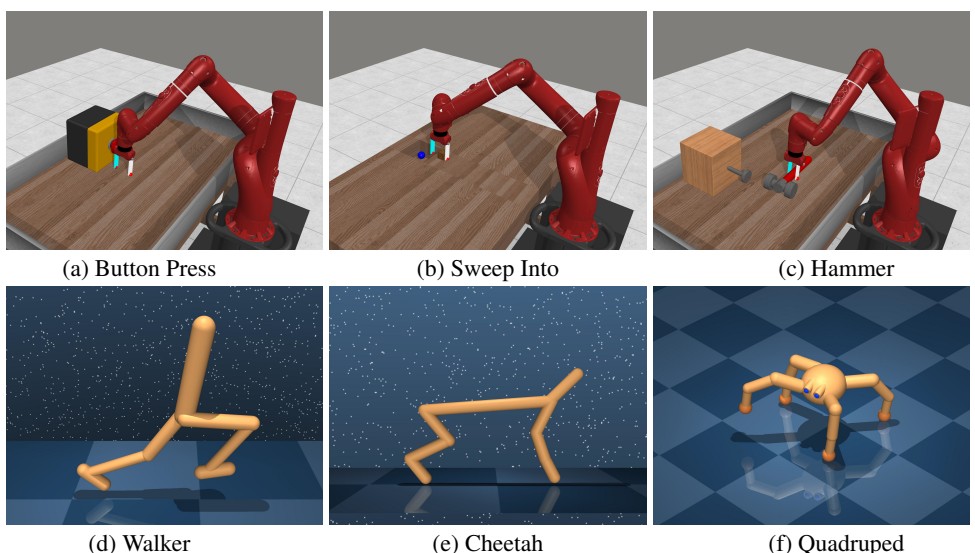

| (a) Button Press | (b) Sweep Into | (c) Hammer |
| (d) Walker | (e) Cheetah | (f) Quadruped |

Figure 8: Six tasks from Meta-world (a-c) and DMControl (d-f).

**Meta-world Tasks:**

○ Button Press: An agent controls a robotic arm to press a button. The button's initial position is randomized.

- ○ Sweep Into: An agent controls a robotic arm to sweep a ball into a hole. The ball's starting position is randomized.
- ○ Hammer: An agent controls a robotic arm to hammer a screw into a wall. The initial positions of both the hammer and the screw are randomized.

**DMControl Tasks:**

- ○ Walker: A planar walker is trained to control its body and walk on the ground.
- ○ Cheetah: A planar biped is trained to control its body and run on the ground.
- ○ Quadruped: A four-legged ant is trained to control its body and limbs, enabling it to crawl on the ground.

## C.2 IMPLEMENTATION DETAILS

For the implementation of baselines, we use their corresponding publicly released repositories that are shown in Table 1. SAC serves as a performance upper bound, because it uses a ground-truth reward function which is unavailable in PbRL settings for training. The detailed hyperparameters of SAC are shown in Table 2. PEBBLE's settings remain consistent with its original implementation, and the specifics are detailed in Table 3. For SURF, RUNE, MRN, and RIME, most hyperparameters are the same as those of PEBBLE and other hyperparameters are detailed in Table 4, 5, 6, and 7, respectively. The total amount of feedback and feedback amount per session in each experimental condition are detailed in Table 8. The reward model comprises an ensemble of three MLPs. Each MLP consists of three layers with 256 hidden units, and the output of the reward model is constrained using the tanh activation function.

Table 1: Source codes of baselines.

| Algorithm | Url |
|---|---|
| SAC, PEBBLE | https://github.com/rll-research/BPref |
| SURF | https://github.com/alinlab/SURF |
| RUNE | https://github.com/rll-research/rune |
| MRN | https://github.com/RyanLiu112/MRN |

Table 2: Hyperparameters of SAC.

| Hyperparameter | Value | Hyperparameter | Value |
|---|---|---|---|
| Number of layers | 2 (DMControl), 3 (Meta-world) | Initial temperature | 0.1 |
| Hidden units per each layer | 1024 (DMControl), 256 (Meta-world) | Optimizer | Adam |
| Learning rate | 0.0005 (Walker), 0.001 (Cheetah) | Critic target update freq | 2 |
| | 0.0001 (Quadruped), 0.0003 (Meta-world) | Critic EMA $\tau$ | 0.005 |
| Batch Size | 1024 (DMControl), 512 (Meta-world) | $(\beta_1, \beta_2)$ | $(0.9, 0.999)$ |
| Steps of unsupervised pre-training | 9000 | Discount $\gamma$ | 0.99 |

Table 3: Hyperparameters of PEBBLE.

| Hyperparameter | Value |
|---|---|
| Segment Length | 50 |
| Learning rate | 0.0005 (Walker, Cheetah), 0.0001 (Quadruped), 0.0003 (Meta-world) |
| Frequency of feedback | 20000 (Walker, Cheetah), 30000 (Quadruped), 5000 (Meta-world) |
| Number of reward functions | 3 |

## D ADDITIONAL EXPERIMENT RESULTS

**Effects of feedback volume on DMControl tasks.** We additionally investigate how the number of noisy feedback influences performance on tasks from DMControl. We conduct experiments on Walker-walk with a fixed error rate $\epsilon = 0.3$ and a varied range of total feedback amount $N \in \{500, 1000, 5000, 10000\}$. As shown in Figure 9, the performance of RIME improves slightly as the

Table 4: Hyperparameters of SURF.

| Hyperparameter | Value |
|---|---|
| Unlabeled batch ratio $\mu$ | 4 |
| Threshold $\tau$ | 0.999 (Cheetah, Sweep Into), 0.99 (others) |
| Loss weight $\lambda$ | 1 |
| Min/Max length of cropped segment | 45/55 |
| Segment length before cropping | 60 |

Table 5: Hyperparameters of RUNE.

| Hyperparameter | Value |
|---|---|
| Initial weight of intrinsic reward $\beta_0$ | 0.05 |
| Decay rate $\rho$* | 0.001 (Walker), 0.0001 (Cheetah, Quadruped, Button Press) |
| | 0.00001 (Sweep Into, Hammer) |

*: Following the instruction of Liang et al. (2021), we carefully tune the hyperparameter $\rho$ in a range of $\rho \in \{0.001, 0.0001, 0.00001\}$ and report the best value for each environment.

Table 6: Hyperparameters of MRN.

| Hyperparameter | Value |
|---|---|
| Bi-level updating frequency $N$ | 5000 (Cheetah, Hammer, Button Press), 1000 (Walker) |
| | 3000 (Quadruped), 10000 (Sweep Into) |

Table 7: Hyperparameters of RIME.

| Hyperparameter | Value |
|---|---|
| Coefficient $\alpha$ in the lower bound $\tau_{\text{lower}}$ | 0.5 |
| Minimum weight $\beta_{\min}$ | 1 |
| Maximum weight $\beta_{\max}$ | 3 |
| Decay rate $k$ | 1/30 (DMControl), 1/300 (Meta-world) |
| Upper bound $\tau_{\text{upper}}$ | $3\ln(10)$ |
| $\delta$ in Equation (9) | $1 \times 10^{-8}$ |
| Steps of unsupervised pre-training | 2000 (Cheetah), 9000 (others) |

Table 8: Feedback amount in each condition.

| Condition | Value* | Condition | Value* |
|---|---|---|---|
| Walker, $\epsilon = 0.1, 0.15$ | 500/50 | Button Press, $\epsilon = 0.1, 0.15$ | 10000/50 |
| Walker, $\epsilon = 0.2, 0.25, 0.3$ | 1000/100 | Button Press, $\epsilon = 0.2, 0.25, 0.3$ | 20000/100 |
| Cheetah, $\epsilon = 0.1, 0.15$ | 500/50 | Sweep Into, $\epsilon = 0.1, 0.15$ | 10000/50 |
| Cheetah, $\epsilon = 0.2, 0.25, 0.3$ | 1000/100 | Sweep Into, $\epsilon = 0.2, 0.25, 0.3$ | 20000/100 |
| Quadruped, $\epsilon = 0.1, 0.15$ | 2000/200 | Hammer, $\epsilon = 0.1, 0.15$ | 20000/100 |
| Quadruped, $\epsilon = 0.2, 0.25, 0.3$ | 4000/400 | Hammer, $\epsilon = 0.2, 0.25$ | 40000/200 |
| | | Hammer, $\epsilon = 0.3$ | 80000/400 |

1. *: Value refers to total amount of feedback / feedback amount per session.

2. $\epsilon$ is the error rate defined in Section 5.1.

number of feedback increases. PEBBLE and RUNE gain marked improvements when the number of feedback is augmented ten to twentyfold, but their performance still lags noticeably behind RIME.

**Effects of hyperparameters of RIME.** We investigate how the hyperparameters of RIME affect the performance under noisy feedback settings. In Figure 10 we plot the learning curves of RIME with different set of hyperparameters: (a) coefficient $\alpha$ in the lower bound $\tau_{\text{lower}}$: $\alpha \in \{0.3, 0.4, 0.5, 0.6\}$, (b) maximum value of $\beta_t$: $\beta_{\max} \in \{1, 3, 5, 10\}$, (c) decay rate $k \in \{0.01, 1/30, 0.06, 0.1\}$, and (d) upper bound of KL divergence $\tau_{\text{upper}} \in \{2\ln(10), 3\ln(10), 4\ln(10)\}$.

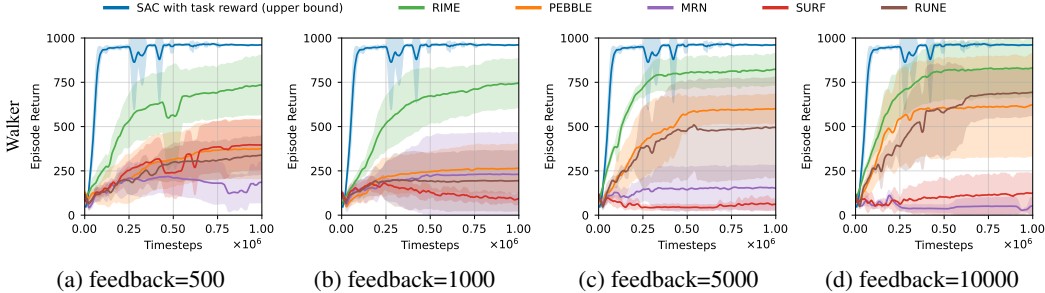

Figure 9: Ablation study on the effects of feedback volume on Walker-walk, where error rate is fixed at 0.3. The results show the mean and standard deviation averaged over five runs.

For the coefficient $\alpha$ in the lower bound $\tau_{\text{lower}}$, we find the theoretical value $\alpha = 0.5$ performs the best. The maximum weight $\beta_{\max}$ and decay rate $k$ control the weight of uncertainty term in the lower bound $\tau_{\text{lower}}$: $\beta_t = \max(\beta_{\min}, \beta_{\max} - kt)$. The combination of $\beta_{\max} = 3$ and $k = 1/30$ also performs optimally. Due to the quite limited feedback amount (1000 feedback) and training epochs for the reward model (around $150 \sim 200$ epochs on Walker-walk), RIME is sensitive to the weight of uncertainty term. If ones try to increase $\beta_{\max}$ to add more tolerance for trustworthy samples in early-stage, we recommend to increase the decay rate $k$ simultaneously so that the value of $\beta_t$ decays to its minimum within about $1/3$ to $1/2$ of the total epochs. For the upper bound $\tau_{\text{upper}}$, although we use $3\ln(10)$ for balanced performance on DMControl tasks, individually fine-tuning $\tau_{\text{upper}}$ can further improve the performance of RIME on the corresponding task, such as using $\tau_{\text{upper}} = 4\ln(10)$ for Walker-walk.

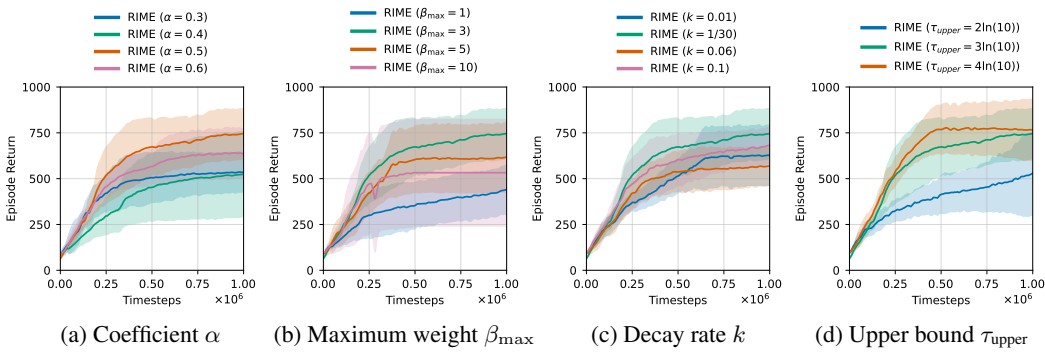

Figure 10: Hyperparameter analysis on Walker-walk using 1000 feedback with $\epsilon = 0.3$. The results show the mean and standard deviation averaged over five runs.

**Effects of different uncertainty terms in the lower bound.** In RIME, we use an auxiliary uncertainty term $\tau_{\text{unc}}$ in the lower bound $\tau_{\text{lower}}$ to accommodate tolerance during the early training stages and in cases of distribution shifts. The standard deviation of the KL divergence, denoted as the KL metric in this section, is employed to discern these cases. Here, we compare this with two other metrics: the disagreement metric and a combination of both, termed as KL + disagreement. The disagreement metric uses the standard deviation of $P_\psi[\sigma^0 \succ \sigma^1]$ across the ensemble of reward models (denoted as $s_P$) to discern cases of distribution shifts: $\tau_{\text{unc}} = \gamma_t \cdot s_P$. Our intuition is that the predictions of the model for OOD data typically vary greatly. Notably, this metric in-

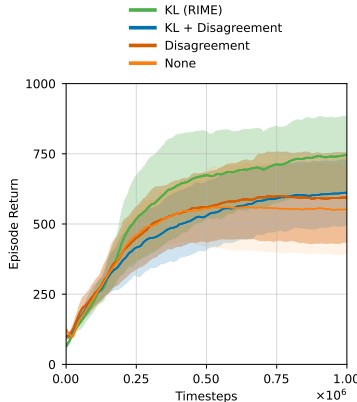

Figure 11: Uncertainty term analysis on Walker-walk using 1000 feedback with $\epsilon = 0.3$, across ten runs.

duces sample-level, rather than buffer-level, thresholds, potentially offering more nuanced threshold control. The combined metric, KL + disagreement, integrates both as $\tau_{\text{unc}} = \beta_t \cdot s_{\text{KL}} + \gamma_t \cdot s_P$.

For reference, we also include a group devoid of any uncertainty term, termed the "None" group. As shown in Figure 11, the KL metric outperforms the other approaches on Walker-walk with an error rate of $\epsilon = 0.3$. This might be because the disagreement metric fluctuates violently at every query times, often leading to excessive trust in new data, which hinders the stabilization of the lower bound.

**Performance of RIME with non-expert human teachers.** Improved robustness should make PbRL more suitable for humans. To investigate this, following Christiano et al. (2017); Lee et al. (2021b); Kim et al. (2022), we conduct a group of experiments with actual non-expert human teachers on Hopper to do backflip. In this experiments, we invite five students in unrelated majors, who are blank to the robotic tasks, to perform online annotation. We only tell them the objective of the task (i.e., teach the agent to do backflip) with nothing else and no further guidance. We utilize their annotations to train RIME and PEBBLE. The feedback amount in total and per session are set to 500 and 100, respectively. Other hyperparameters are kept the same with those of RIME on Walker-walk.

We employ a hand-crafted reward function designed by experts (Christiano et al., 2017) as the ground-truth scripted teacher. We found that compared to ground-truth preferences, our non-expert annotation error rate reached nearly 40%. Therefore, we additionally investigate the performance of both algorithms with scripted teacher at error rate $\epsilon = 0.4$. The results are shown in Fig. 12. We find that RIME significantly outperforms PEBBLE when learning from actual non-expert human teachers and successfully performs consecutive backflips using 500 non-expert feedback, as shown in Figure 13. Furthermore, both algorithms performed worse with script-prone teachers than human teachers at the same error rate, suggesting that the "Mistake" model in (Lee et al., 2021a) may be more difficult than real noise.

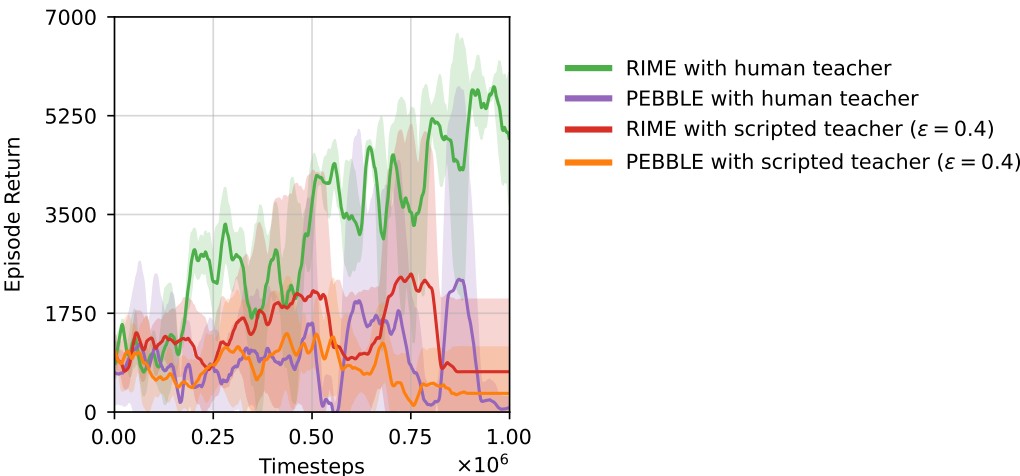

Figure 12: Performance on Hopper with non-expert human teachers. The results show the mean and standard deviation averaged over five runs.

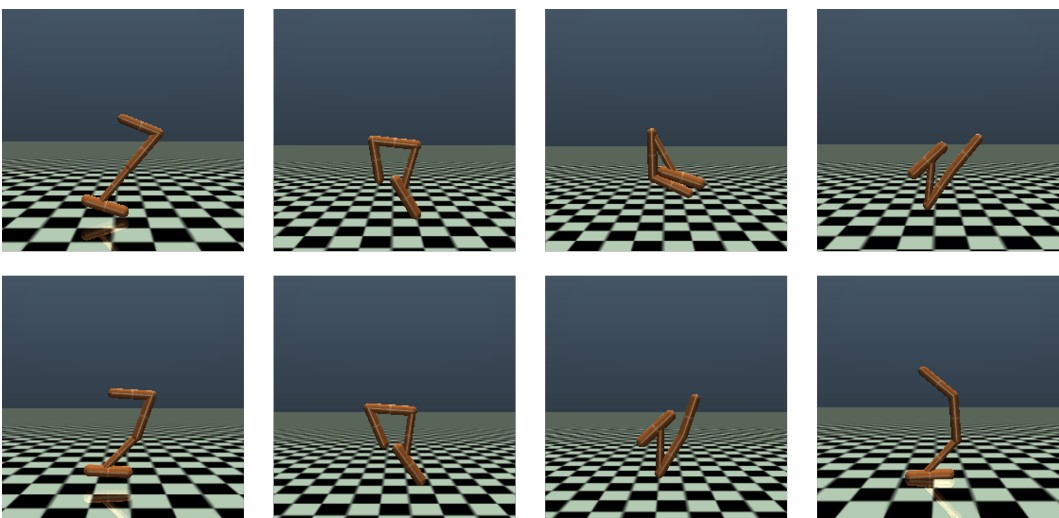

Figure 13: Novel behaviours trained using feedback from non-expert human teachers. RIME successfully executes continuous backflips using 500 non-expert feedback.

