# OpenReview forum: "RIME: Robust Preference-based Reinforcement Learning with Noisy Human Preferences"
_ICLR.cc/2024/Conference — ICLR 2024 Conference Withdrawn Submission_

### Official Review · Reviewer_vzpZ · 2023-10-15

**Soundness:** 3 good
**Presentation:** 3 good
**Contribution:** 2 fair
**Rating:** 6
**Confidence:** 3

**Summary:**

The authors propose a discriminator and a warm-up strategy to solve noisy labeling in RLHF.

**Strengths:**

1. Noisy labeling is an important problem in RLHF.
2. The authors propose two techniques a discriminator and a warm-up technique to solve the problem.
3. The authors also provide ablation studies for the two techniques

**Weaknesses:**

1. For the discriminator part, the bound in the theory is not well analyzed. How large is the constant in the squared term $O(\rho^2)$? Does it affect the value of the bound if the constant is large?  Would it be possible to quantitatively visualize some of the cases and the corresponding bound values in experiments?
2. There are some hyperparamters $\alpha, \beta$ in the bound for the discriminator. These hyperparameters are somehow vague and make the connection between the theory and practive loose. A concrete analysis of these hyperparameters and how they affect the final results should be included.

**Questions:**

See the weakness section

---

> ### Author Response · Authors · 2023-11-14
>
> Dear Reviewer vzpZ,
>
> We sincerely appreciate your valuable and insightful comments. We found them extremely helpful for improving our manuscript. We address each comment in detail, one by one below.
>
> ---
>
> **Q1. Analysis of the Constant in $\mathcal{O}(\rho^2)$**
>
> **A1.** We appreciate your attention to the details regarding the constant in the squared term $\mathcal{O}(\rho^2)$. The theoretical value of this constant is $-\frac{1}{24}$. In our experiments, the bound $\rho$ for cross-entropy loss in clean data typically varies between 0.2 and 1.0. This range implies that, relative to the term $-\ln \rho + \frac{\rho}{2}$, the contribution of the squared term $\mathcal{O}(\rho^2)$ to the lower bound $\tau_\text{lower}$ is indeed minor. Consequently, it is reasonable to omit this term in our analysis and experiments for clarity and simplicity without significantly impacting the overall results and conclusions.
>
> ------
>
> **Q2. Hyperparameter Analysis for $\alpha$ and $\beta$**
>
> **A2.** We thank you for your query regarding the hyperparameter analysis. Detailed analysis of these parameters is presented in the appendix of our paper, specifically in Figure 10. For the coefficient $\alpha$ in the lower bound $\tau_\text{lower}$, we find that a theoretical value of $\alpha=0.5$ yields optimal performance. Additionally, the hyperparameter $\beta_{\max}$ and the decay rate $k$ control the weight $\beta_t$ of the uncertainty term in $\tau_\text{lower}$, defined as $\beta_t=\max(\beta_{\min}, \beta_{\max}-kt)$. Our experiments demonstrate that a combination of $\beta_{\max}=3$ and $k=1/30$ is most effective. We encourage a review of these findings in our appendix for a more comprehensive understanding.

---

### Official Review · Reviewer_z4ZA · 2023-10-31

**Soundness:** 3 good
**Presentation:** 3 good
**Contribution:** 2 fair
**Rating:** 8
**Confidence:** 4

**Summary:**

This paper introduces RIME, an algorithm to increase the robustness of Preference-based Reinforcement Learning (PbRL). RIME adds a preference discriminator used to filter out (or even flip) really noise labels. Additionally, RIME introduces a warm-start method to better transition from the unsupervised exploration phase of PbRL methods to the proper reward training phase.

Experiments show RIME outperforming a thorough selection of baseline when the labelling error is high.

**Strengths:**

* Both theoretical contributions (bounding the divergence of a preference discriminator and warm-start) are interesting and useful for the community. In particular, the soft transition between the unsupervised phase and the reward training phase that warm-start induces, seems to me more logical than simply reseting the reward.
* The experimental section is very thorough, comparing (and beating) most recent baselines when the noise of the mistake labeller is high. Equally the ablations serve to understand the behaviour of RIME, in particular the fact that RIME does not show negative effects when training on clean labels.

**Weaknesses:**

* _W1_: The paper does not feature a user study with actual human labellers. This is particularly relevant as ultimately the gains in robustness are meant to make PbRL more usable by humans.
* _W2_: There is no composition of RIME with other methods, but the contributions of RIME are surely applicable to SURF or RUNE.
* _W3_: RIME uses a uniform sampling schedule rather than an uncertainty based one to select the initial trajectories to query. But such choice is not explained, nor ablated, and other methods (like PEBBLE) use the uncertainty-based sampling.

**[[Post-rebuttal update]]**

The authors added extra experiments with actual human labellers, combined with other baselines, as well as further sampling schedules, clearing most of my weaknesses.

The remaining minor weaknesses are that RIME does not seem to always take advantage of the improvements proposed in other PbRL methods such as MRN or RUNE, and that the analyses so far have not included standard deviation across 5 or 10 runs.

**Questions:**

* _Q1_: Is the preference discriminator another module? Or is it embedded into the learnt reward function? The text seems to indicate the former, but Figure 1 points to the latter. Either way, please include a description of how the discriminator is operationalised.
* _Q2_: Do you have an intuition as to why RIME performs approximately the same as PEBBLE in DMC's cheetah?
* _Q3_: At what value of $\epsilon$ does RIME stop working? At the reported $\epsilon = 0.3$ most tasks continue to perform acceptably.
* _Q4_: [Less important] Could you add an analysis of what proportion of labels get suppressed /flipped under different $\epsilon$, tasks, and decay thresholds? This would be useful to better understand the role of label discrimination.

**Nitpicks and suggestions (will not affect rating)**

* In the literature review, consider quantifying the sentence "huge amount of preference labels"
* In section 4.1, consider including the proof of Theorem 1 (at least for y=0) in the main manuscript, it is an important part of the paper.
* In section 4.1, explicitly state that $\alpha$ must be $\le 0.5$, otherwise it will not be a lower-bound of the KL divergence.
* In section 4.1, explain that the linear decay schedule for $\beta$ will be denoted as $k$ further down the manuscript.
* In section 5, I found the explicit research questions useful. Could you link add references to the sections with the actual experiments?
* Could you add some videos/screenshots of the final performance of RIME with some tasks (eg. Cheetah)?
* I found Figure 5, particularly hard to parse. I think a table with final performance and all the ablations would be easier to interpret.
* Consider separating Figure 6 into two half-page figures. Subfigures a) and b) have very little to do with subfigures c) and d).
* In section 5.2, I am not sure how "RIME improves robustness whilst maintaining feedback efficiency" follows "Given that it takes only a few human minutes to provide 1000 preference labels". Also note that more difficult tasks (which are ultimately the goal of PbRL) may well require more time to annotate. Think about rephrasing that sentence?


**[[Post-rebuttal update]]**

The authors addressed all the major questions above and updated the manuscript accordingly. See the discussion for more details.

---

> ### Author Response · Authors · 2023-11-21
>
> Dear Reviewer z4ZA,
>
> Thank you for your valuable and insightful feedback on our manuscript. We have carefully considered your comments and revised our paper accordingly. These revisions are highlighted in blue for your convenience. Below, we address each of your comments in detail.
>
> ---
>
> **W1. User study with actual human labellers**
>
> **A1.** We appreciate your suggestion regarding the inclusion of a user study with actual human labellers. We add a group of case study with actual human labellers. In this experiment, we invite five students in unrelated majors, who are blank to the robotic tasks, to perform online annotation. We only tell them the objective of the task (i.e., teach the agent to do backflip on Hopper) with nothing else and no further guidance. We utilize their annotations to train RIME and PEBBLE and compare the performance.
>
> The experimental hyperparameters on Hopper are shown in Table 1, and the results are shown in Table 2. Hyperparameters not specified in Table 1 (e.g., learning rate, batch size, feedback frequence, etc.) are consistent with those used for RIME on Walker-walk.
>
> We have augmented the appendix of our paper to comprehensively detail this experiment and results. Additionally, we have included in the supplementary materials the **code for online human annotation** and **GIFs** illustrating the results of training based on human preferences. Our findings indicate that RIME successfully executes continuous backflips using 500 non-expert feedback.
>
> Table 1: Experiment details with actual human labellers on Hopper
>
> | Hyperparameter              | Value |
> | --------------------------- | ----- |
> | Total amount of feedback    | $500$ |
> | Feedback amount per session | $100$ |
>
> Table 2: Experiment results with actual human labellers on Hopper, across five humans
>
> | Algorithm | True episode return |
> | --------- | ------------------- |
> | RIME      | $5813$              |
> | PEBBLE    | $2197$              |
>
> ---
>
> **W2. Combine RIME with other baselines**
>
> **A2.** Thank you for pointing this out. Technically, RIME is applicable to SURF and RUNE. However, our contributions are different from them. They intend to improve feedback-efficiency for PbRL, aiming at maximizing the expected return with few number of preference queries, while we intend to improve robustness for PbRL, enhancing performance in noisy feedback settings. Due to the mismatch of contributions, we have not applied RIME to SURF or RUNE. However, we recognize the potential interest and challenge in enhancing robustness with limited preferences in PbRL and have earmarked this as an area for future research.
>
> ---
>
> **W3. Analysis of sampling schedule**
>
> **A3.** We appreciate your suggestion to delve deeper into the analysis of sampling schedules. Prior to our main experiments, we conducted preliminary tests to choose the sampling schedule. In noisy feedback environments, uniform sampling emerged as the most effective strategy. We hypothesize that more complex sampling methods, such as uncertainty-based or entropy-based, depend heavily on the output of the reward model. In noisy settings, a biased reward model could lead to misleading queries, thereby hindering algorithm performance. We present a comparative study of three sampling schedules on Walker-walk with an error rate of 0.3, conducted over ten runs, in Table 3. Our final draft will include additional results as part of an "ablation study on sampling schedule."
>
> Table 3: Performance of three algorithms on Walker-walk ($\epsilon=0.3$) with different sampling schedules
>
> |        | Uniform  | Uncertainty-based | Entropy-based |
> | ------ | -------- | ----------------- | ------------- |
> | RIME   | $738.71$ | $671.08$          | $694.85$      |
> | PEBBLE | $278.33$ | $257.79$          | $276.63$      |
> | MRN    | $248.62$ | $204.59$          | $265.44$      |
>
> ---
>
> **Q1. Description of how the discriminator is operationalised**
>
> **A4.** We apologize for any lack of clarity in our initial description of the discriminator's role. The discriminator is embedded into the learnt reward function. It is a manually defined function with no extra learnable parameters, using the current reward function $\hat{r}_\psi$ to determine whether the input sample is corrupted.  For a given segment pair $(\sigma^0,\sigma^1)$ and the associated annotated label $\tilde{y}$, the discriminator computes the predicted preferences based on Equation (1). It then calculates the KL divergence between the predicted label and the annotated label $\tilde{y}$. This divergence is compared against two thresholds to identify and filter out noisy preferences in accordance with Equations (6) and (7).

---

> ### Author Response · Authors · 2023-11-21
>
> **Q2. Explanation of the result in DMC's cheetah**
>
> **A5.**  We appreciate your interest in the distinctive results observed in DMC's cheetah environment. It was noted that PEBBLE exhibits robust performance, particularly in the initial stages of training, surpassing that of RIME. Our investigations suggest that minimizing the unsupervised pre-training steps improves RIME's performance in Cheetah. Therefore, we hypothesize that the unsupervised pre-training potentially leads to a suboptimal policy in Cheetah, making it challenging for the agent to progress optimally. This issue appears to impact RIME more significantly than PEBBLE, owing to RIME inheriting all parameters (policy, value, and reward network), which results in comparable performance between the two algorithms in the cheetah environment.
>
> ---
>
> **Q3. The limit of RIME with respect to $\epsilon$**
>
> **A6.** Thank you for pointing this out. We conducted additional experiments on the Walker-walk and Button-press environments with varying error rates $\epsilon$ set at 35%, 40%, and 45%, over five runs. The results, presented in Table 4, indicate that RIME fails with 45% noisy data on Walker-walk and fails with 40% noisy data on Button-press.
>
> Table 4: Performance of RIME with different error rates
>
> | **Env/Noise** | metric              | 35%      | 40%      | 45%      |
> | ------------- | ------------------- | -------- | -------- | -------- |
> | Walker-walk   | True episode return | $653.43$ | $476.36$ | $159.22$ |
> | Button-press  | Success rate        | $65$%    | $15$%    | $7.5$%   |
>
> ---
>
> **Q4. More analysis of label discrimination**
>
> **A7.** We appreciate your interest in a deeper analysis of label discrimination under varying conditions. We present the proportion of labels get suppressed / flipped under different $\epsilon$, tasks, and  thresholds after convergence in the following tables. Let $\mathcal{D}$ be the original noisy preference dataset. $\mathcal{D}_t$ and $\mathcal{D}_f$ are defined in Equations (6) and (7) respectively. $|\cdot|$ represents the number of elements in the set. The column named "accuracy of $\mathcal{D}_i$" ($i\in \{t,f\}$) means the number of clean samples in $\mathcal{D}_i$ / $|\mathcal{D}_i|$.
>
> Table 5: More information about label discrimination with different $\epsilon$ on **Walker-walk**
>
> | $\epsilon$ | $\tau_{lower}$ | $\mid\mathcal{D}_t\mid/\mid\mathcal{D}\mid$ | $\mid\mathcal{D}_f\mid/\mid\mathcal{D}\mid$ | accuracy of $\mathcal{D}_t$ | accuracy of $\mathcal{D}_f$ |
> | ---------- | ------------------- | --------------------------------------- | --------------------------------------- | --------------------------- | --------------------------- |
> | $0.1$      | $0.924$             | $93.2$%                                 | $4.6$%                                  | $92.70$%                    | $60.87$%                    |
> | $0.15$     | $0.977$             | $91.2$%                                 | $4.8$%                                  | $90.13$%                    | $58.33$%                    |
> | $0.2$      | $0.948$             | $87.6$%                                 | $4.6$%                                  | $85.27$%                    | $63.04$%                    |
> | $0.25$     | $0.965$             | $85.8$%                                 | $3.5$%                                  | $81.35$%                    | $62.86$%                    |
> | $0.3$      | $0.972$             | $82.8$%                                 | $4.1$%                                  | $73.43$%                    | $68.29$%                    |
>
> Table 6: More information about label discrimination with different $\epsilon$ on **Button-press**
>
> | $\epsilon$ | $\tau_{lower}$ | $\mid\mathcal{D}_t\mid/\mid\mathcal{D}\mid$ | $\mid\mathcal{D}_f\mid/\mid\mathcal{D}\mid$ | accuracy of $\mathcal{D}_t$ | accuracy of $\mathcal{D}_f$ |
> | ---------- | ------------------- | --------------------------------------- | --------------------------------------- | --------------------------- | --------------------------- |
> | $0.1$      | $0.711$             | $90.29$%                                | $4.19$%                                 | $94.41$%                    | $68.97$%                    |
> | $0.15$     | $0.712$             | $87.26$%                                | $5.62$%                                 | $91.46$%                    | $75.80$%                    |
> | $0.2$      | $0.710$             | $78.66$%                                | $7.22$%                                 | $87.42$%                    | $74.50$%                    |
> | $0.25$     | $0.710$             | $80.78$%                                | $11.11$%                                | $84.69$%                    | $77.04$%                    |
> | $0.3$      | $0.711$             | $73.92$%                                | $16.07$%                                | $82.34$%                    | $75.08$%                    |

---

> ### Author Response · Authors · 2023-11-21
>
> **Q5. Editorial comments/clarification**
>
> **A8.** Thank you very much for your detailed comments. We will incorporate the suggested editorial comments and clarifications in the final draft.

---

> > ### Comment · Reviewer_z4ZA · 2023-11-22
> > **Response to rebuttal**
> >
> > Thank you for your detailed rebuttal and the additional experiments you have conducted. In what follows I address individually each of the issues discussed.
> >
> > -----------
> >
> > **W1. User study with actual human labellers**
> >
> > Thank you for running the additional experiments. The results and attached GIFs are very interesting.
> >
> > It is not completely clear to me how did you train the model with user preferences. Did you collect different preference sets for PEBBLE and RIME? If not, how did you deal with the fact that the preferences came from off-policy trajectories. Did the preferences from different users get mixed?
> >
> > Additionally, it would be interesting to know how long did it take each annotator to label the preferences.
> >
> > Clarifying these questions would be helpful for other practitioners wanting to emulate these experiments.
> >
> >
> > **W2. Combine RIME with other baselines**
> >
> > I am not convinced that maximising sample efficiency, and increasing robustness are mismatched. It is conceivable that, for instances, RUNE's uncertainty selection can improve select more informative trajectories that RIME could filter and improve.
> >
> > But even if other methods are not compatible with RIME, this is important information for practitioners in order to decide whether to implement RIME.
> >
> > **W3. Analysis of sampling schedule**
> >
> > The results are reassuring and the hypothesis makes sense, it is encouraging to see that RIME still beats PEBBLE under uncertainty-sampling. Thank you.
> >
> > **Q1. Description of how the discriminator is operationalised**
> >
> > Thank you for the clarification. In retrospect, my confusion came from not realising that $P_\psi(\sigma^0, \sigma^1)$ (ie the predicted label) _is_ the output of the reward estimator $\hat{r}_\psi$, which I tend to think of as the _predicted reward_.
> >
> > It may be worth clarifying this in the manuscript, in case other readers do not realise this either.
> >
> >
> > **Q2. Explanation of the result in DMC's cheetah**
> >
> > Thank you for the explanation and for the hypothesis. Unfortunately, the following does not make sense to me: "This issue appears to impact RIME more significantly than PEBBLE, owing to RIME inheriting all parameters (policy, value, and reward network), which results in comparable performance between the two algorithms in the cheetah environment." Why would inheriting the hyper-parameters of PEBBLE impact negatively RIME? This is presumably not the case for all other experiments.
> >
> > As an alternative hypothesis, could it be that RIME is more dependent on a good estimation of the reward, and that by the time it transitions to the objective in eq. (8), the reward has overfit to the unsupervised reward objective?
> >
> > If this is the case, it would be advisable to highlight this limitation in the manuscript.
> >
> >
> > **Q3. The limit of RIME with respect to epsilon.**
> >
> > Thank you for running this experiment. It is interesting the RIME can deal with such values of $\epsilon$. Please add this ablation to the manuscript as an appendix.
> >
> > **Q4. More analysis of label discrimination**
> >
> > Thank you for running this experiment, it does help better understand RIME. $\mathcal{D}_t$'s behaviour makes sense the higher the ratio of noisy samples, the more difficult it is to establish an adequate KL-divergence lower-bound, and thus the lower the ratio of clean samples that are selected.
> >
> > However, I am intrigued by the behaviour of Df. In walker-walk, the accuracy seems pretty low. Perhaps a larger $\tau_{upper}$ threshold would perform better? This also helps explains why in Figures 5a) and 5b) the purple line is close to the green line. Yet in Button-press, though $\tau_{upper}$ is presumably fixed, the accuracy of $\mathcal{D}_f$ goes up as the $\epsilon$ increases. Do you have any hypothesis as to why?
> >
> > Needless to say, I recommend this analysis be included in the manuscript as an appendix. I would also urge the authors to add the value of $\tau_{upper}$ to the table, $3\ln(10)$ according to the manuscript.
> >
> > -------------
> >
> > I am aware there is very little time till the end of the discussion, but I would be very grateful if authors could respond to the follow-up questions in _W1_, _Q2_, and _Q4_

---

> > > ### Author Response · Authors · 2023-11-22
> > > **Response to z4ZA (Part I)**
> > >
> > > We sincerely thanks for your serious and responsible reply! Due to the time limited, we address the following-up questions *W1, W2, Q2, and Q4* one by one below.
> > >
> > > ---
> > >
> > > **W1. Details about the user study with actual human labelers**
> > >
> > > **A1.** Our experiment adopts an online paradigm consistent with PEBBLE's pipeline, where agent training alternates with reward model training. When it is the timestep to collect preferences (post-agent training and pre-reward training), the training program generates segment pairs $\{(\sigma^0_i,\sigma^1_i)\}_{i=1}^N$, saving each segment in GIF format. The program then pauses, awaiting the input of human preferences. At this juncture, human labelers engage, reading the paired segments in GIF format and labelling their preferences. Subsequently, this annotated data is fed into the training program for reward model training.
> > >
> > > To ensure a fair comparison between RIME and PEBBLE, for one human labeler, we start the training programs for RIME and for PEBBLE simultaneously. When both training programs are waiting for inputting preferences, we collate all gif pairs from both RIME and PEBBLE and shuffle the order. Then the human labeler starts working. Therefore, from the labeler's perspective, he does not know which algorithm the currently labeled segment pair comes from and just focuses on labelling according to his preference. The labeled data is then automatically directed to the respective training programs.  We conduct this experiment  parallelly on each of the five labelers, thus preferences from different users do not get mixed.
> > >
> > > In the Hopper task, the labeling process itself requires approximately 20 minutes, not accounting for waiting time. However, due to the online annotation, labelers experience downtime while waiting for agent training and GIF pair generation. Consequently, considering all factors, the total time commitment for labeling amounts to about 1 hour per annotator.
> > >
> > > ---
> > >
> > > **W2. Combine RIME with other baselines**
> > >
> > > **A2.** We appreciate your insight to combine RIME with other baselines and especially your reasonable example. Therefore, we conduct experiments on this topic and presents the results in the following table. We find that combining RIME with RUNE or MRN indeed promotes significant improvement. The implementation of RIME based on PEBBLE still shows the most robust performance among the experimental algorithms. However, SURF + RIME shows no significant improvement. We hypothesize that the preference data augmented from corrupted samples seriously affects the algorithm performance.
> > >
> > > Table 1: Performance of the combination of RIME with other baselines on Walker-walk with $\epsilon=0.3$
> > >
> > > | Algorithm | True episode return |
> > > | --------- | ------------------- |
> > > | PEBBLE    | $278.33$            |
> > > | SURF      | $121.48$            |
> > > | RUNE      | $219.37$            |
> > > | MRN       | $248.42$            |
> > > | RIME      | $738.71$            |
> > > | SURF+RIME | $153.29$            |
> > > | RUNE+RIME | $695.71$            |
> > > | MRN+RIME  | $638.26$            |

---

> > > ### Author Response · Authors · 2023-11-22
> > > **Response to z4ZA (Part II)**
> > >
> > > **Q2. Explanation of the result in DMC's cheetah**
> > >
> > > **A3.** We are sorry that we do not understand the meaning of the following sentence: "Why would inheriting the hyper-parameters of PEBBLE impact negatively RIME? This is presumably not the case for all other experiments." We feel there may be a misunderstanding between us. We restate our hypothesis in detail.
> > >
> > > We hypothesize that the unsupervised pre-training potentially leads to a suboptimal policy in Cheetah, making it challenging for the agent to progress optimally. This issue appears to impact RIME more significantly than PEBBLE, owing to RIME **inheriting all parameters gained from pre-training stage (i.e., policy, value, and reward network parameters, not the hyper-parameters of PEBBLE**). Moreover, due to the performance gap during transition shown in Fig. 2, PEBBLE may forgets the pre-trained policy and thus avoid the suboptimal problem in pre-trained policy.
> > >
> > > As for your hypothesis: "could it be that RIME is more dependent on a good estimation of the reward, and that by the time it transitions to the objective in eq. (8), the reward has overfit to the unsupervised reward objective?" To investigate this, we present  more information of label discrimination in the first three reward training sessions in the following table. As shown in Table 2, the performance metrics associated with $\tau_\text{lower}$ and $\tau_\text{upper}$ are quite normal. Therefore, the hypothesis of overfitting which further results in incorrect sample selection and thus biases the objective in Eq. (8) probably is not true.
> > >
> > > Table 2: More information of label discrimination in the first three reward training sessions on Cheetah with $\epsilon=0.1$
> > >
> > > | Training times of reward model | Current amount of feedback | Value of $\tau_\text{lower}$ | $\mid\mathcal{D}_t\mid/\mid\mathcal{D}\mid$ | $\mid\mathcal{D}_f\mid/\mid\mathcal{D}\mid$ | Accuracy of $\mathcal{D}_t$ | Accuracy of $\mathcal{D}_f$ |
> > > | ------------------------------ | -------------------------- | ---------------------------- | ------------------------------------------- | ------------------------------------------- | --------------------------- | --------------------------- |
> > > | 1                              | 50                         | 3.31                         | 74%                                         | 8%                                          | 92.50%                      | 60%                         |
> > > | 2                              | 100                        | 2.77                         | 87%                                         | 4%                                          | 91.21%                      | 75%                         |
> > > | 3                              | 150                        | 1.11                         | 91%                                         | 1.3%                                        | 93.84%                      | 50%                         |
> > > | 4                              | 200                        | 1.02                         | 90.5%                                       | 2%                                          | 92.82%                      | 50%                         |
> > > | 5                              | 250                        | 0.93                         | 92.6%                                       | 1.6%                                        | 92.47%                      | 50%                         |

---

> > > ### Author Response · Authors · 2023-11-22
> > > **Response to z4ZA (Part III)**
> > >
> > > **Q4.  More analysis of $\tau_\text{upper}$ in label discrimination**
> > >
> > > **A4.** **Larger $\tau_\text{upper}$:** As shown in Fig. 10(d) in appendix, increasing the $\tau_\text{upper}$ to $4\ln(10)$ improves the performance of RIME on Walker-walk. Although we use $3\ln(10)$ for balanced performance on DMControl tasks, individually fine-tuning $\tau_\text{upper}$ can further improve the performance of RIME on the corresponding task, such as using $\tau_\text{upper} = 4 \ln(10)$ for Walker-walk. But note that the benefit from adjusting $\tau_\text{upper}$ may not be significant, as indicated by the minor disparity between the green and orange lines in Figure 10(d).
> > >
> > > **Accuracy of $\mathcal{D}_f$:** In Table 6 in our first round of response, there is no obvious trend that the accuracy of $\mathcal{D}_f$ (the last column in Table 6) increases as $\epsilon$ increases when $\epsilon \in\{0.15,0.2,0.25,0.3\}$. Only the 68.97% accuracy of $\mathcal{D}_f$ in $\epsilon=0.1$ seems abnormal, which is lower than that in other conditions. But this result seems consistent with the result shown in Fig. 5(c), which demonstrates the limited role of upper bound on Button-press with $\epsilon  =0.1$. As for the reason, we assume that since lower bound plays a great role in this condition (shown in Fig. 5(c)), resulting higher accuracy of $\mathcal{D}_t$, there are more corrupted samples in the sample set filtered by lower bound (i.e., the set $\mathcal{D}-\mathcal{D}_t$). Thus the accuracy of $\mathcal{D}-\mathcal{D}_t$ is lower. As the label flipping with upper bound is operated on the set $\mathcal{D}-\mathcal{D}_t$, the accuracy of $\mathcal{D}_f$ is lower.

---

> > > > ### Comment · Reviewer_z4ZA · 2023-11-23
> > > >
> > > > Overall, I thank the authors for thoroughly answering my questions and addressing the weaknesses (at the last minute too!). The paper is stronger now than when I reviewed it, so consequently I will raise the rating.
> > > >
> > > > As a last minute suggestion, I urge authors to include standard deviations and tables to their manuscript, these are key to establish how whether an algorithm is predictable. Apologies for not having raised this concern earlier.
> > > >
> > > > Regarding the individual issues we were discussing:
> > > >
> > > > -------
> > > >
> > > > **W1** Thank you for clarifying the protocol. This is a reasonable way to obtain human preferences and validates that RIME does indeed outperform PEBBLE with actual human preferences. Once more, for the next version of the paper, do not forget to include the user-study protocol as well as the standard deviation.
> > > >
> > > > **W2** Thank you for running these experiments. I note that SURF+RIME is still a 26% improvement over SURF. But I agree that SURF's augmentations are likely at fault. It is a pity that neither MRN+RIME nor RUNE+RIME beat PEBBLE+RIME, but clearly this setting is challenging for both MRN and RUNE. Better integration with other PbRL algorithms can be left for future work. And yet again, this table would benefit from standard deviations.
> > > >
> > > > **Q2** Thank you for the clarification. Indeed, I had misinterpreted your response and ignored the fact that PEBBLE resetd the reward network while RIME does not. Your explanation makes sense, and clearly my hypothesis was not correct (thanks for running the ablation nonetheless).
> > > >
> > > >
> > > > **Q4** Thanks for the detailed response. I had missed Fig 10(d), but your explanation makes sense.  Regarding the accuracy of Df, your analysis seems right, I had missed the fact that $\tau_{upper}$ operates on the Dt set. I suggest further explaining the role of $\tau_{upper}$ (following this discussion) to the manuscript.

---

### Official Review · Reviewer_dSpC · 2023-11-01

**Soundness:** 1 poor
**Presentation:** 2 fair
**Contribution:** 2 fair
**Rating:** 3
**Confidence:** 5

**Summary:**

The work attempts to make PbRL algorithms like PEBBLE more robust by proposing a sample-selection method, i.e. to reject feedback samples obtained from a scripted oracle (stand in for human in the loop) if the proposed method finds it too “noisy“. Additionally, they argue in favor of using a ”warm-start“ method for reward learning. For empirical evaluation they consider the case of ”mistake-oracle“ as defined in a prior work and evaluate on 6 tasks across 2 domains which have also been used in past PbRL works.

**Strengths:**

The work is straight-forward add-on method over an existing backbone PbRL algorithm and is easy to use with PbRL algorithms with PEBBLE like training paradigm.

The authors do a good job at highlighting their method  and provide enough background / information for readers. It is easy to follow (and can invite constructive criticism).

I like the general theme of the work, which is to study noisy human preferences and I agree that it is an important topic for the PbRL / RLHF community (however I have concerns : see Weaknesses/Questions).

**Weaknesses:**

1. The title of the paper says “noisy human preferences” however there is no study or analysis with real humans in the loop. Is there a reason to believe that humans provide feedback as “mistake oracle” does? As I recall, even the authors of BPref agreed that the suggested oracles in their work may not truly reflect human preference. Do the authors have some experiments with actual humans interacting with the RIME framework giving their preferences on domains like in some past PbRL works?
2. The work has been done entirely in the context of a specific “mistake oracle” as defined in a prior work (BPref [1]). I think this relates to previous point, while the works like PEBBLE / SURF / RUNE / MRN were done in the context of perfect teachers and that BPref was done to promote better scripted teachers : I am surprised that authors did not investigate other scripted models for noisy teachers. For example, Noise could have been conditioned on state / trajectory etc. While the claims of the paper are on “robustness” and that their study on “mistake-oracle“ is a good starting point, I do not see how improvements along a single noisy oracle justifies generality of the approach.
3. I also feel that the experiments are quite limited and generally present information that do not answer the main claim on the robustness.
    1. Limited Domains : For example, authors only show results on 6 tasks across 2 domains, where there is no significant performance improvements on Walker/Cheetah over baseline PEBBLE.
    2. Limited baselines for handling noise : I would have considered experiments on the lines of Fig 6c, 6d as the most important. Figures 3, 4 generally shows that existing PbRL methods are not robust to noise which is a valid point to make. However, it is figure 6c/d which shows why RIME is a superior method to alternate solutions against label noise. The authors only have two baselines and report results on two domains with specific noise error values. I was not able to find more results in the appendix.
4. What is the overlap of the presented method on “sample selection” based robustness to noise and existing literature. While there may not be works in PbRL directly tackling noise, it is unclear from section 2 Related Work what are the existing approaches for robustness that are used in Machine Learning and how they may resemble the proposed work. This is especially important as PbRL reward learning is posed as a classification problem, and there is a large body of works on robustness in classification.
5. How is warm-start something that is related to human preferences or reward learning or robustness to noise? It seems that warm-start can be applied to a learning problem and may potentially provide performance improvements. Authors can argue against if they want, but as I understand warm-start is a general add-on technique that may provide more reasonable initialization to an approximator (there are some preliminary works within the context of PbRL as well [2]). Additionally, Fig 5a,b suggests that the the performance boost with the sample-selection strategy is very limited compared to baseline PEBBLE. Infact RIME (only tau lower/upper) is worse for eps=0.1. Do the authors have results for PEBBLE  + Warm start only?


[1] Lee, K., Smith, L., Dragan, A., & Abbeel, P. (2021). B-pref: Benchmarking preference-based reinforcement learning. arXiv preprint arXiv:2111.03026.

[2] Verma, M., & Kambhampati, S. (2023). Data Driven Reward Initialization for Preference based Reinforcement Learning. arXiv preprint arXiv:2302.08733.

**Questions:**

1. Can the authors provide some insights into the rejection rate of RIME? That is, how many samples eventually are not considered by the algorithm during training. It is unclear how many “noisy” samples still go through the training.
2. The authors state that they use the uniform sampling scheme for query selection. Is this choice for all the baselines and RIME? If so, PEBBLE proposed more advanced methods of query sampling like disagreement and uncertainty based which can generate better queries. In general, I would argue that query selection can have a big impact on noise (if actual humans are providing feedback). Even in the current setup, methods like PEBBLE have favored more advanced query sampling strategies.
3. Can the authors report reward recovery results (true episode returns) on the Metaworld tasks for which they have reported success rate? Or comment on the difference in reward recovery?
4. Why have the authors chosen different feedback schedules for different noise values? (as in Table 8 which I think should be moved to the main text. Otherwise figures 3/4 can become misleading. For example, for Quadruped it appears that at eps = 0.2 the performance is better than at eps = 0.15, but it is because the authors provide double the total feedback.)

I would also request the authors to respond to the points in “Weakness” section.

---

> ### Author Response · Authors · 2023-11-22
> **Response to dSpC (Part I)**
>
> Dear Reviewer dSpC,
>
> We sincerely appreciate your valuable and insightful comments. We found them extremely helpful for improving our manuscript. We have updated our revision based on your comments and colored by blue. We address each comment in detail, one by one below.
>
> ---
>
> **W1. Experiments with actual humans**
>
> **A1.** We appreciate your suggestion regarding the inclusion of a user study with actual human labellers. We add a group of case study with actual human labellers. In this experiment, we invite five students in unrelated majors, who are blank to the robotic tasks, to perform online annotation. We only tell them the objective of the task (i.e., teach the agent to do backflip on Hopper) with nothing else and no further guidance. We utilize their annotations to train RIME and PEBBLE and compare the performance.
>
> The experimental hyperparameters on Hopper are shown in Table 1, and the results are shown in Table 2. Hyperparameters not specified in Table 1 (e.g., learning rate, batch size, feedback frequence, etc.) are consistent with those used for RIME on Walker-walk.
>
> We have **augmented the appendix** of our paper to comprehensively detail this experiment and results. Additionally, we have included in the supplementary materials the **code for online human annotation** and **GIFs** illustrating the results of training based on human preferences. With additional content in the updated appendix, we find that RIME significantly outperforms PEBBLE when learning from actual non-expert human teachers and successfully performs consecutive backflips using 500 non-expert feedback. Moreover, the experimental results verified one of our conjectures, that is, improvements made under the "mistake oracle" are still effective in real noise. Therefore, the robust algorithm under the "mistake oracle" model can be generalized to real situations.
>
> Table 1: Experiment details with actual human labellers on Hopper
>
> | Hyperparameter              | Value |
> | --------------------------- | ----- |
> | Total amount of feedback    | $500$ |
> | Feedback amount per session | $100$ |
>
> Table 2: Experiment results with actual human labellers on Hopper, across five humans
>
> | Algorithm | True episode return |
> | --------- | ------------------- |
> | RIME      | $5813$              |
> | PEBBLE    | $2197$              |
>
> ---
>
> **W2. Experiments on more scripted models for noisy teachers**
>
> **A2.** Thank you for pointing this out. Bpref proposed four 0-1 labeled noisy teacher models: "Mistake", "Skip", "Equal", and "Myopic". They found PEBBLE achieved approximately efficient performance on "Skip" and "Equal" compared with perfect teacher model (Oracle), while PEBBLE often suffer from poor performance on "Mistake" teacher model. This led us to select the "Mistake" model for our research, as it presents the most challenging scenario. We hypothesize that the solutions for this case can generalize to other easier noisy cases. To validate this, we conduct experiments of RIME on other noisy cases (i.e., "Skip", "Equal", and "Myopic") and present the results in Table 3. All the hyperparameters of noise models are kept the same with BPref. Our final draft will include additional results as part of an "ablation study on other noise cases."
>
> As shown in Table 3,  the "Skip" and "Equal" noise even improves the performance of both algorithms. Furthermore, the "Myopic" teacher model, which introduces noise based on the timestep, is effectively managed by RIME in a manner akin to the "Mistake" model. This demonstrates that RIME, initially developed for the "Mistake" noise, exhibits a capacity for generalization to other noise model scenarios.
>
> Table 3: Experiments on other noise teacher models with 500 feedback on Walker-walk
>
> | Noise model                         | Algorithm | True episode return |
> | ----------------------------------- | --------- | ------------------- |
> | Oracle                              | RIME      | $958.02$            |
> | Oracle                              | PEBBLE    | $884.97$            |
> | Mistake $(\epsilon=0.1)$            | RIME      | $912.42$            |
> | Mistake $(\epsilon=0.1)$            | PEBBLE    | $767.34$            |
> | Skip $(\epsilon_\text{adapt}=0.1)$  | RIME      | $964.35$            |
> | Skip $(\epsilon_\text{adapt}=0.1)$  | PEBBLE    | $910.43$            |
> | Equal $(\epsilon_\text{adapt}=0.1)$ | RIME      | $962.91$            |
> | Equal $(\epsilon_\text{adapt}=0.1)$ | PEBBLE    | $922.46$            |
> | Myopic $(\gamma=0.9)$               | RIME      | $920.64$            |
> | Myopic $(\gamma=0.9)$               | PEBBLE    | $762.53$            |

---

> > ### Comment · Reviewer_dSpC · 2023-11-23
> > **Response to authors rebuttal**
> >
> > I appreciate the authors' response and additional experiments to make their claims.
> >
> > While it is impressive that the authors could put together several pieces of experiments, it adds some additional concern :
> > 1. The existing experiments were incomplete in justifying the benefits of RIME. (more oracles, human study, more baselines).
> > 2. The authors did the experiments during the short rebuttal phase which may have causes unintended errors. [But I will stay optimistic on this point].
> >
> >
> > **On User Study** : I have similar concerns as Reviewer : z4ZA as raised in their response. That is, whether the data was collected in an online manner (i.e. RIME builds up the trajectory buffer which is used to query the participants) or whether the queries were made by populating the trajectory buffer via random walk. Additionally, authors report that the noise is 40% which seems extremely high, but possible. In any case, PEBBLE reports that they were able to train a Hopper-Backflip using 50 queries. While they do not report the quality (i.e. noise level metric) on their collected queries, it is considerably less than 500 queries taken by the authors. Further, it is unclear whether the natural-noise (or noise level shown by real human) would be same across the study in PEBBLE and the one that authors report.
> >
> >
> > **Experiments on More Teachers ** : I appreciate the authors efforts in conducting experiments on additional oracles. I would have liked a larger empirical evaluation on these lines beyond walker-walk domain [however, I do not mean that authors must do it during the rebuttal phase, bur rather such experiments should have been an important part of their empirical analysis to begin with].
> >
> > Additionally, I feel the authors have an opportunity to contribute other oracles focussed on errors that humans in the loop may make. As I had pointed out, mistake can be random (i.e. the current mistake oracle), and can also depend on state or trajectory features. Some investigation on the relationship between the noise humans in the loop tend to exhibit and the oracle they consider is an important precursor to proposing algorithms which can takle mistake-oracle based noise. In prior works, mistake oracle is seen as a loose substitute to provide general trends of the PbRL algorithms when the feedback is noisy, whereas the claim of the current work is to be robust against noise - in which case the types of noise matter more.
> >
> > **Limited Baselines** I appreciate the authors initiative in identifying better baselines. The main proposal from SURF was improved efficiency in the same setting as PEBBLE and other PbRL. RIME on the other hand proposes robustness while maintaining efficiency, which highlights why 6c, 6d figures are more important. As their results indicate, other baselines like MAE come close to RIME performance, but is shown on a single domain. [I would like to reiterate, I would have appreciated an extensive evaluation on this line, but I do not expect the authors to complete such experiments during the rebuttal phase.]
> >
> > **warm start** : I would like to thank the authors for posting the additional result on PEBBLE + WS. I feel warm start is a useful technique that can be used for making any learning system more "robust". The authors make a valid point that it is a useful part of their overall contribution towards making their system more robust, but it seems more of an ad-hoc technique than specific to learning from human preferences. The authors discussion on this is very useful in clarifying this.
> >
> > ** Rejection Rate ** : The authors present an interesting result that sampling technique does not seem to have a very large impact when epsilon is high. Previous works have shown the importance of choosing a sampling technique, which seems to not hold when the there is mistake-like noise in feedback. This is an important result that authors can stress on if this observation holds beyond one domain.
> >
> > Finally, I would like to appreciate the authors efforts and in producing several experiments during the short rebuttal phase (which can be difficult). The additional experiments shows the need for an updated evaluation section, and the added baselines should be discussed in related work on robustness in ML.
> >
> > I believe once the authors extend these experiments, they can build a stronger revised submission.

---

> ### Author Response · Authors · 2023-11-22
> **Response to dSpC (Part II)**
>
> **W3. Limited domains and limited baselines for handling noise**
>
> **A3.** **Limited domains:** Following most of the PbRL works adopting PEBBLE's pipeline (PEBBLE, SURF, RUNE, MRN), all of the works conduct experiments on these two domains: DMC and Metaworld. Additionally, RIME does exceeds baselines by a large margin on Walker-walk with $\epsilon=0.1, 0.2, 0.3$.  As for the Cheetah, we find that minimizing the unsupervised pre-training steps improves the performance of RIME in Cheetah. Thus we hypothesize that the unsupervised pre-training potentially leads to a suboptimal policy, making it challenging for the agent to progress optimally in the cheetah environment. This issue appears to impact RIME more significantly than PEBBLE, owing to RIME inheriting all parameters (policy, value, and reward network), which results in comparable performance between the two algorithms in the cheetah environment.
>
> **Limited baselines for handling noise:** Our experimental design and paper writing mainly refer to SURF, as both RIME and SURF pioneer certain aspects within PbRL – SURF in efficiency with few prefereces and RIME in robustness. Therefore, following SURF, our primary experiments (illustrated in Figures 3 and 4) contrast with PbRL baselines, while our ablation studies (Figures 6(c) and 6(d)) compare with baseline + two specific methods addressing specific issues.
>
> In response to your constructive suggestion, here we compare RIME with other two classical robust training methods: label smoothing regularization[1] and mean absolute error (MAE) loss function [2], on Walker-walk with $\epsilon=0.3$. For label smoothing, we use $\hat{y}=0.9\tilde{y}+0.05$ instead of $\tilde{y}$ for the cross-entropy computation. For MAE, we use the following loss function instead of cross-entropy loss. We will add more results on different environments and more error rates in our final draft.
> $$
> \mathcal{L}(P_\psi,\tilde{y})=\frac1N\sum_{i=1}^N|{P_\psi}_i-\tilde{y}_i|
> $$
> Table 4: Comparison with other sample selection methods for robust training on Walker-walk with $\epsilon=0.3$
>
> | Robust training method                               | True episode return |
> | ---------------------------------------------------- | ------------------- |
> | PEBBLE (no robust training)                          | $473.38$            |
> | RIME (sample selection)                              | $738.71$            |
> | Fixed threshold (sample selection)                   | $551.30$            |
> | Adaptive denoising training (ADT) (sample selection) | $575.99$            |
> | Label smoothing (robust regularization)              | $482.31$            |
> | Mean absolute error (MAE) (robust loss function)     | $631.42$            |
>
> [1] Lukasik M, Bhojanapalli S, Menon A, et al. Does label smoothing mitigate label noise? International Conference on Machine Learning. PMLR, 2020: 6448-6458.
>
> [2] Ghosh A, Kumar H, Sastry P S. Robust loss functions under label noise for deep neural networks. Proceedings of the AAAI conference on artificial intelligence. 2017, 31(1).
>
> ---
>
> **W4. Overlap of the presented method on “sample selection” based robustness to noise and existing literature**
>
> **A4.** We appreciate your insight into the overlap of our method with existing literature on robust training. Existing works in robust training can be divided into four categories: robust architecture, robust regularization, robust loss function, and sample selection methods.
>
> 1. Robust architecture: adding a noise adaptation layer at the top of an underlying DNN to learn label transition process or developing a dedicated architecture to reliably support more diverse types of label noise.
> 2. Robust regularization: enforcing a DNN to overfit less to false-labeled examples explicitly or implicitly.
> 3. Robust loss function: improving the loss function or adjusting the loss value according to the confidence of a given loss (or label) by loss correction, loss reweighting, or label refurbishment.
> 4. Sample selection: identifying true-labeled examples from noisy training data. Our approach can be situated within the sample selection category.
>
> There are two significant challenges in directly applying these methods to PbRL. Firstly, they require large amount of samples, typically on the order of millions. Secondly, they assume that the samples are equally distributed. However, during the RL training process, the state distribution encountered is shifting. This dynamic nature makes it challenging to differentiate new data from noise.

---

> ### Author Response · Authors · 2023-11-22
> **Response to dSpC (Part III)**
>
> **W5. How is warm-start something that is related to human preferences or reward learning or robustness to noise?**
>
> **A5.** Thank you for highlighting this aspect. Warm-start plays a multifaceted role in our approach.
>
> - In terms of noise robustness, it offers a well-calibrated initialization for the discriminator, mitigating the accumulation of errors due to incorrect selection.
> - With respect to reward learning, warm-start bridges the performance gap observed during the transition from pre-training to online training. This not only conserves the number of queries required in the early stages of learning but also allocates more queries for deeper, more effective learning later in the process.
> - Furthermore, the recognition of a performance gap during the transition from pretraining to online training in PbRL is a noteworthy contribution of our work. Our experiments demonstrate that addressing this issue through warm-start significantly benefits both the effectiveness and robustness of the learning process.
>
> As for the ablation study on component analysis, Fig. 5(a) shows the crucial role of warm-start in feedback-limited and small error rate case. Fig. 5(b) demonstrates that removing warm-start or removing $\tau_\text{lower}$ are all fatal to the performance, and only the combination of these components can achieve the overall success of RIME. Fig. 5(c) and 5(d) further shows the limited role of warm-start in feedback-adequated cases (20000 feedback). We present the result with PEBBLE + warm-start in the following table.
>
> Table 5: Performance of PEBBLE + warm-start (WS) on Walker-walk with different error rates
>
> | Environment | Error rate | Algorithm | Metric   |
> | ----------- | ---------- | --------- | -------- |
> | Walker-walk | 0.1        | RIME      | $912.42$ |
> | Walker-walk | 0.1        | PEBBLE+WS | $850.12$ |
> | Walker-walk | 0.1        | PEBBLE    | $767.34$ |
> | Walker-walk | 0.3        | RIME      | $738.71$ |
> | Walker-walk | 0.3        | PEBBLE+WS | $424.31$ |
> | Walker-walk | 0.3        | PEBBLE    | $278.33$ |

---

> ### Author Response · Authors · 2023-11-22
> **Response to dSpC (Part IV)**
>
> **Q1. Provide some insights into the rejection rate of RIME**
>
> **A6.** We appreciate your interest in a deeper analysis of label discrimination under varying conditions. We present the proportion of labels get suppressed / flipped under different $\epsilon$, tasks, and  thresholds after convergence in the following tables. Let $\mathcal{D}$ be the original noisy preference dataset. $\mathcal{D}_t$ and $\mathcal{D}_f$ are defined in Equations (6) and (7) respectively. $|\cdot|$ represents the number of elements in the set. The column named "accuracy of $\mathcal{D}_i$" ($i\in \{t,f\}$) means the number of clean samples in $\mathcal{D}_i$ / $|\mathcal{D}_i|$.
>
> Table 5: More information about label discrimination with different $\epsilon$ on **Walker-walk**
>
> | $\epsilon$ | $\tau_{lower}$ | $\mid\mathcal{D}_t\mid/\mid\mathcal{D}\mid$ | $\mid\mathcal{D}_f\mid/\mid\mathcal{D}\mid$ | accuracy of $\mathcal{D}_t$ | accuracy of $\mathcal{D}_f$ |
> | ---------- | -------------- | ------------------------------------------- | ------------------------------------------- | --------------------------- | --------------------------- |
> | $0.1$      | $0.924$        | $93.2$%                                     | $4.6$%                                      | $92.70$%                    | $60.87$%                    |
> | $0.15$     | $0.977$        | $91.2$%                                     | $4.8$%                                      | $90.13$%                    | $58.33$%                    |
> | $0.2$      | $0.948$        | $87.6$%                                     | $4.6$%                                      | $85.27$%                    | $63.04$%                    |
> | $0.25$     | $0.965$        | $85.8$%                                     | $3.5$%                                      | $81.35$%                    | $62.86$%                    |
> | $0.3$      | $0.972$        | $82.8$%                                     | $4.1$%                                      | $73.43$%                    | $68.29$%                    |
>
> Table 6: More information about label discrimination with different $\epsilon$ on **Button-press**
>
> | $\epsilon$ | $\tau_{lower}$ | $\mid\mathcal{D}_t\mid/\mid\mathcal{D}\mid$ | $\mid\mathcal{D}_f\mid/\mid\mathcal{D}\mid$ | accuracy of $\mathcal{D}_t$ | accuracy of $\mathcal{D}_f$ |
> | ---------- | -------------- | ------------------------------------------- | ------------------------------------------- | --------------------------- | --------------------------- |
> | $0.1$      | $0.711$        | $90.29$%                                    | $4.19$%                                     | $94.41$%                    | $68.97$%                    |
> | $0.15$     | $0.712$        | $87.26$%                                    | $5.62$%                                     | $91.46$%                    | $75.80$%                    |
> | $0.2$      | $0.710$        | $78.66$%                                    | $7.22$%                                     | $87.42$%                    | $74.50$%                    |
> | $0.25$     | $0.710$        | $80.78$%                                    | $11.11$%                                    | $84.69$%                    | $77.04$%                    |
> | $0.3$      | $0.711$        | $73.92$%                                    | $16.07$%                                    | $82.34$%                    | $75.08$%                    |
>
> ---
>
> **Q2. Analysis of sampling scheme**
>
> **A7.** We appreciate your suggestion to delve deeper into the analysis of sampling schedules. Prior to our main experiments, we conducted preliminary tests to choose the sampling schedule. In noisy feedback environments, uniform sampling emerged as the most effective strategy. We hypothesize that more complex sampling methods, such as uncertainty-based or entropy-based, depend heavily on the output of the reward model. In noisy settings, a biased reward model could lead to misleading queries, thereby hindering algorithm performance. We present a comparative study of three sampling schedules on Walker-walk with an error rate of 0.3, conducted over ten runs, in Table 7. Our final draft will include additional results as part of an "ablation study on sampling schedule."
>
> Table 7: Performance of three algorithms on Walker-walk ($\epsilon=0.3$) with different sampling schedules
>
> |        | Uniform  | Uncertainty-based | Entropy-based |
> | ------ | -------- | ----------------- | ------------- |
> | RIME   | $738.71$ | $671.08$          | $694.85$      |
> | PEBBLE | $278.33$ | $257.79$          | $276.63$      |
> | MRN    | $248.62$ | $204.59$          | $265.44$      |

---

> ### Author Response · Authors · 2023-11-22
> **Response to dSpC (Part V)**
>
> **Q3. True episode returns on the Metaworld tasks**
>
> **A8.** We acknowledge your interest in the true episode returns for the Metaworld tasks. Following prior works (PEBBLE/SURF/RUNE/MRN), we report the success rate on Metaworld because the ultimate goal of control tasks is to successfully complete tasks (reaching expeteced states) rather than maximize predefined returns. However, in alignment with your request, we have included the results of true episode returns on Metaworld in Table 8.
>
> Table 8: True episode returns on Meta-world tasks of RIME across ten runs
>
> | Environment  | Error rate | Total feedback amount | True episode return |
> | ------------ | ---------- | --------------------- | ------------------- |
> | Button-press | $0.1$      | $10000$               | $3292.03$           |
> | Button-press | $0.15$     | $10000$               | $3197.68$           |
> | Button-press | $0.2$      | $20000$               | $3335.99$           |
> | Button-press | $0.25$     | $20000$               | $2840.01$           |
> | Button-press | $0.3$      | $20000$               | $2499.76$           |
> | Sweep-into   | $0.1$      | $10000$               | $2232.19$           |
> | Sweep-into   | $0.15$     | $10000$               | $2075.12$           |
> | Sweep-into   | $0.2$      | $20000$               | $2597.90$           |
> | Sweep-into   | $0.25$     | $20000$               | $2126.73$           |
> | Sweep-into   | $0.3$      | $20000$               | $1135.09$           |
> | Hammer       | $0.1$      | $20000$               | $2847.89$           |
> | Hammer       | $0.15$     | $20000$               | $2759.56$           |
> | Hammer       | $0.2$      | $40000$               | $2636.23$           |
> | Hammer       | $0.25$     | $40000$               | $1590.47$           |
> | Hammer       | $0.3$      | $80000$               | $2762.82$           |
>
> ---
>
> **Q4. Why have the authors chosen different feedback schedules for different noise values?**
>
> **A9.** Thank you for pointing this out. We start our main experiments (Fig. 3 and 4) from $\epsilon=0.3$ with the chosen feedback amount. As degrading the error rate $\epsilon$, we find RIME can exceeds baselines with fewer feedback amount. Therefore, we degrade the feedback amount simultaneously for feedback-efficient and saving computing resources in small error rate cases.

---

> ### Author Response · Authors · 2023-11-22
> **A mild reminder**
>
> We sincerely thank you for your constructive comments. And we would like to know whether we have addressed your concerns. If so, might you be able to update your rating to reveal this? Our goal is to ensure that our responses and modifications meet your expectations and enhance the quality of our work. We are eager to continue this dialogue and are available for any further discussion that may be helpful.